# Widespread temporal niche partitioning in an adaptive radiation of cichlid fishes

Annika L. A. Nichols [1,6], Maxwell E. R. Shafer [1,5,6] ✉, Adrian Indermaur[2], Attila Rüegg [2], Rita Gonzalez-Dominguez[1], Milan Malinsky [2,3], Carolin Sommer-Trembo[2,4], Laura Fritschi[2], Amelia Mesich [5], Ayasha Abdalla-Wyse [5], Walter Salzburger [2] & Alexander F. Schier [1]

The partitioning of ecological niches is a fundamental component of species diversification in adaptive radiations. However, it is currently unknown if and how such bursts of organismal diversity are influenced by temporal niche partitioning, wherein species avoid competition by being active or sleeping during different time windows. Here we address this question through profiling temporal activity patterns in the exceptionally diverse fauna of cichlid fishes from the African Lake Tanganyika. By integrating week-long longitudinal behavioural recordings of over 500 individuals from 60 species with eco-morphological and genomic information, we provide two lines of evidence that temporal niche partitioning occurs in this massive adaptive radiation. First, Tanganyikan cichlids exhibit all known circadian temporal activity patterns (diurnal, nocturnal, crepuscular and cathemeral) and display substantial interspecific variation in daily amounts of locomotion. Second, many species with similar habitat and diet niches occupy distinct temporal niches. Moreover, our results suggest that shifts between diurnal and nocturnal activity patterns are facilitated by a crepuscular intermediate state. Genome-wide association studies indicate that the genetics underlying activity patterns is complex, with different clades associated with different combinations of variants. The identified variants were not associated with core circadian clock genes but with genes implicated in synapse function. These observations indicate that temporal niche partitioning may have contributed to adaptive radiation in cichlids and that many genes are associated with the diversity and evolution of temporal activity patterns.

Adaptive radiations are characterized by rapid species diversification as a consequence of niche specialization[1–3]. For example, the beaks of Darwin's finches are highly specialized for different diets[3], the varied limbs of anole lizards allow access to different sections of their tropical forest habitat[4] and the diverse body and jaw shapes of African cichlid fishes match their habitats and diets[5]. Ecological niches can also be of temporal nature[6]. For example, species can specialize to be more active during certain time windows, such as the day, the night or twilight periods.

While such cases of temporal niche partitioning have been well documented between distantly related species that coexist in sympatry[7–9], it is largely unknown whether variation in circadian temporal activity patterns can contribute to niche specialization in the context of adaptive radiations. Additionally, little is known about the molecular or genetic mechanisms that underlie different circadian activity patterns.

Cichlid fishes (Cichlidae) are one of the most species-rich families of vertebrates[10]. They commonly diversify through adaptive radiation,

as seen across their geographic distribution and especially in the East African Great Lakes[5,11,12]. At approximately 9–12 million years of age, Lake Tanganyika is the oldest of the East African Great Lakes. This lake harbours the most diverse adaptive radiation of cichlid fishes, displaying huge disparity in morphology, ecology, behaviour and genetics[12]. Anecdotal observations have suggested that cichlid species are mostly diurnal and rely heavily on visual systems and colouration patterns associated with daylight[13]. However, there are reports that some species of cichlids are nocturnal[14] or harbour traits often associated with a nocturnal lifestyle, such as large eyes and an expanded lateral line[13,15,16]. These observations raise the possibility that temporal niche partitioning also occurs in cichlid adaptive radiations, with eco-morphologically similar species occupying distinct temporal activity patterns. Here we test this hypothesis by examining temporal activity patterns and their genetic underpinnings in the cichlid fishes of Lake Tanganyika.

## Results

### Activity patterns are highly diverse in Tanganyikan cichlids

To investigate the temporal activity patterns in the cichlid fish fauna of Lake Tanganyika, we developed a behavioural tracking paradigm to follow hundreds of individually housed fish over week-long periods in a reductionist and controlled laboratory setting, or 'common garden'[17–20]. We collected data for 60 species from the adaptive radiation of cichlid fishes in Lake Tanganyika, including representatives from 9 out of the 12 tribes, and covering all trophic levels and all major diet guilds[5] (Fig. 1a). We examined both the largest and smallest cichlids in the world (*Boulengerochromis microlepis* (abbreviated as Boumic) and *Neolamprologus multifasciatus* (Neomul); see Supplementary Data 1 for a list of species names and abbreviations) and species from all major biotypes in that lake, resulting in a phylogenetically, ecologically and morphologically diverse and representative set of species. In our experiments, we continuously tracked each individual fish for 6 days and 6 nights at high temporal resolution (10 Hz) within tanks separated by mesh dividers (Fig. 1b). This setup allowed fish to interact with each other through visual and olfactory cues, while facilitating robust tracking of individuals. Up to 14 adult individuals per species were tracked (average of nine individuals, with a range from 2 to 14; Supplementary Data 1). Tracking by custom Python code recorded the position and speed of each fish.

We uncovered a remarkable diversity of activity patterns across these closely related species (Fig. 1, Extended Data Figs. 1 and 2 and Supplementary Data 2). Patterns ranged from diurnal (for example, in *Neolamprologus buescheri* (Neobue)) to nocturnal activity (for example, in *Neolamprologus toae* (Neotoa)) to peaks in activity at dawn and dusk (for example, in *Neolamprologus pulcher* (Neopul)) to no strong preference in daily activity (for example, in *Enantiopus melanogenys* (Enamel)) (Fig. 1c). Our results are largely consistent with anecdotal observations from the wild available for some of the species. For example, *N. toae* is known to feed on insect larvae during the night, has very large eyes relative to its body size and an expanded lateral line system[13]. In comparison, its congener *N. buescheri* has a smaller relative eye size than *N. toae*, is colourful and is known to be aggressive and active during the day[13]. Our results also match activity patterns reported anecdotally in the literature based on observations of feeding activities in the lake, including nocturnal activity for *N. toae*, *Neolamprologus tretocephalus*[21] and *Aulonocranus dewindti*[22], and crepuscular/diurnal activity for *Neolamprologus brichardi*, *Neolamprologus savoryi*[23] and *N. pulcher*[24]. Moreover, quantification of the activity patterns for three species in their home tanks, which include conspecifics and environmental enrichment, were consistent with the results of our reductionist experimental approach (Extended Data Fig. 3). These examples suggest that the activity patterns measured in the lab match known physiological adaptations and temporal activity patterns in nature.

To determine the major axes of variation in behavioural activity patterns across species, we performed dimensionality reduction using principal component analysis (PCA) of their 30-min-binned average daily swimming speed, with species as features. This approach revealed two components that explained 72% of the variation of the daily activity patterns (PC1 explained 45% and PC2 explained 27% of variance, while PC3 only explained 8%) (Fig. 1d,e). The PC1 axis separated day time points (when lights were on) from night time points (when lights were off) (Fig. 1d), demonstrating that PC1 corresponds largely to day–night differences in activity preferences (Fig. 1f). PC2 represented variation in time points associated with changing light conditions (dawn and dusk, Fig. 1d), revealing the crepuscular (dawn/dusk) preferences of activity (Fig. 1f). PCs 3 to 10 had smaller contributions over the 24-hour period, explaining only 0.5–7.5% of the variation in swimming speed. This analysis revealed that a species' daily activity patterns have a diurnal/nocturnal (PC1) and a crepuscular (PC2) component, and that a species' temporal activity pattern can be measured by their loadings for each. For example, hierarchical clustering based on the species loadings of PC1 and PC2 showed that the species form three distinct groups, which we designated as diurnal (negative PC1 loadings), crepuscular (high PC2 loadings, PC1 loadings near 0) and nocturnal (positive PC1 loadings) (Fig. 1g). In addition, several species exhibited high variability in activity patterns in our experiment (Extended Data Fig. 1). This included species such as *E. melanogenys* (Enamel) and *Neolamprologus cygnatus* (Neocyg), some of which also displayed weak preferences for diurnal, nocturnal and crepuscular periods (PC1 and PC2 loadings near 0). These species probably lack temporal preferences for activity and occupy a cathemeral lifestyle. Therefore, we added a cathemeral class to species with very high variability (Fig. 1g; Methods). Interestingly, this analysis shows that a species' preference for diurnal or nocturnal activity is not mutually exclusive with its preference for crepuscular peaks of activity. For example, *Tropheus moorii* (Tromoo) is diurnal (negative PC1 loading), but also crepuscular (positive PC2 loading), whereas *Astatotilapia burtoni* (Astbur) is diurnal (negative PC1 loading), but lacks crepuscular peaks of activity (low PC2 loading) (Fig. 1g,h). Together these results suggest that the Tanganyikan cichlids display all known activity patterns (diurnal, nocturnal, crepuscular, cathemeral). In addition, our results suggest that crepuscularity is not mutually exclusive with diurnality or nocturnality.

### Total rest is highly diverse in Tanganyikan cichlids

Given the diversity in the timing of activity across species, we next asked whether cichlids display variation in their total amounts of activity or inactivity per day. Inactivity can be used as a proxy for sleep in fishes; for example, it is commonly used in the diurnal zebrafish, which exhibit short bouts of reduced motility associated with higher arousal thresholds predominantly during the night[17,25,26]. Cichlids also displayed periods of inactivity. We used our high-resolution tracking data to quantify the total daily amount of consistent inactivity periods (less than 5% of movement in a sliding 60-s window), which we called 'rest' (Fig. 1i; Methods). As fish size did not correlate with fish speed, we used an absolute threshold for movement (Extended Data Fig. 4a,b). The majority of species rested near the substrate in the bottom quarter of the tank, including *E. melanogenys* (Enamel) and *Telmatochromis salzburgeri*[27] (Telsal) (Extended Data Fig. 4c,d). Some exceptions were those with more pelagic lifestyles such as *Cyprichromis coloratus* (Cypcol), which tended to rest within the water column, and *Paracyprichromis nigripinnis* (Pcynig), which tended to rest off the bottom but next to the mesh walls of the arena (Extended Data Fig. 4e), with both species adopting a more vertical posture during rest compared to active periods (Extended Data Fig. 4f). We found extensive variation in total rest between species (Fig. 1i). For example, some species rested for up to 18 h per day (for example, *Altolamprologus sp.* 'compressiceps shell' (Altshe)), while others displayed less than 3 h of rest per day (for example, *Ophthalmotilapia boops* (Ophboo) and *Limnotilapia dardennii* (Limdar)). These results demonstrated that the range in daily total rest in cichlids resembles the range known across all mammal species

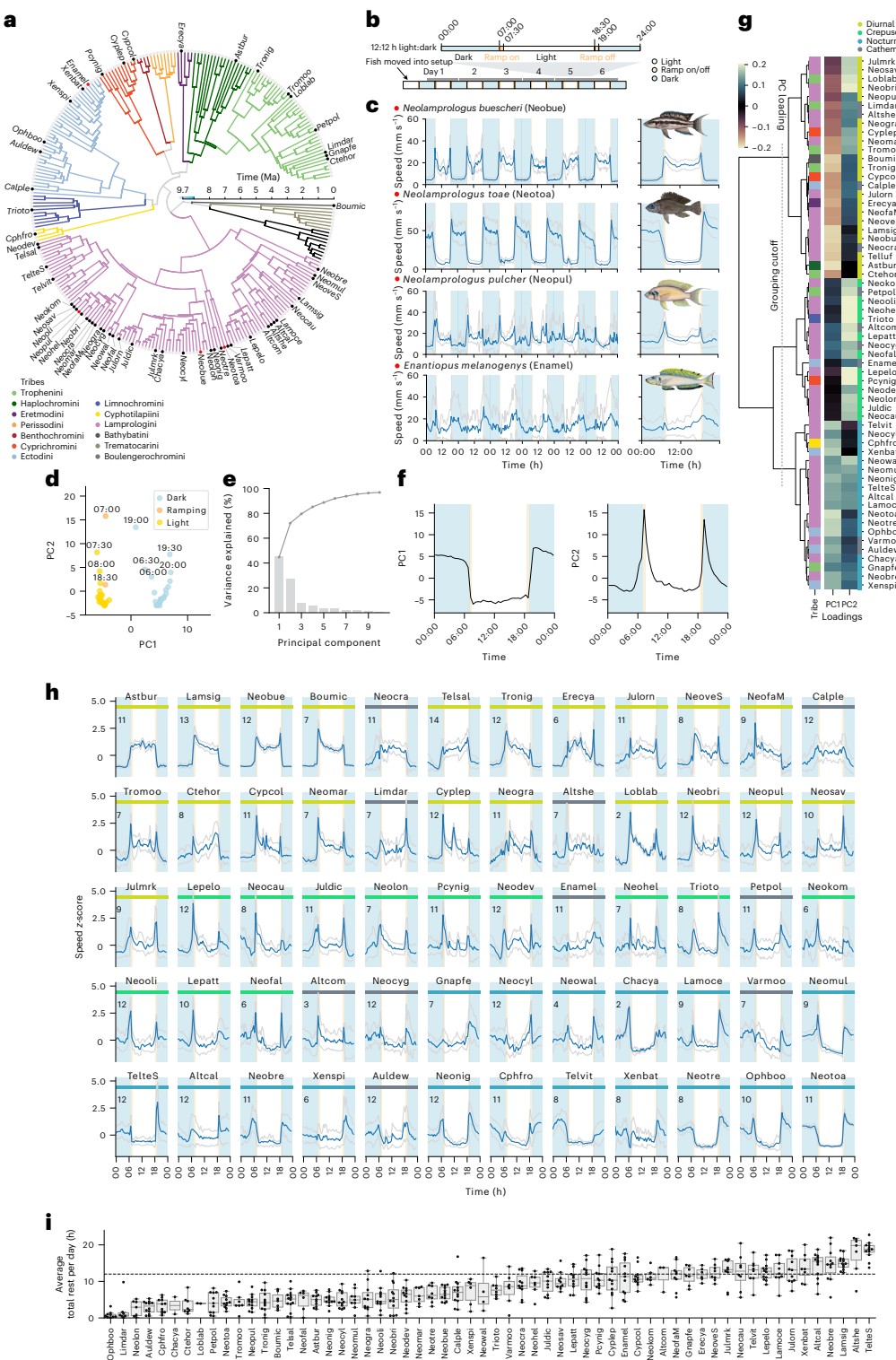

**Fig. 1 | Cichlids occupy all known temporal activity niches and display extensive variation in total rest across species. a–c**, Time-calibrated phylogenetic tree of cichlid species from the Lake Tanganyikan radiation with branches coloured according to tribe (**a**). Labelled species were included in our behavioural screen. Red dots indicate example species shown in **c**. **b**, Schematic of the timeline and light cycle for the behavioural assays. **c**, The weekly and daily average speed traces (mean ± s.d.) of four example species. **d**, PCA analysis of the daily speed averages across the 60 species separates out the 30-min time bins by light state. **e**, The variance explained by the first ten principal components. The overlaid line shows the cumulative sum of variance explained. **f**, Plot of PC1 and PC2 values. **g**, The clustered loadings of PC1 and PC2 are plotted along with the tribes (same colour key as in **a**), and diel guilds. Dotted line indicates the cutoff for determining temporal activity pattern groupings. **h**, The z-score normalized average daily speed traces for each of the 60 cichlid species assayed ordered by PC1 loadings. Coloured bar indicates the temporal activity pattern of the species (same colour key as in **g**). Numbers of animals assayed for each species are shown in the top-left corner in each plot. **i**, Average daily total rest for each species; each dot shows the average for one individual. Species names are abbreviated using a six-letter code following ref. 5 (Supplementary Data 1). The centre line of each box represents the median, the box represents the 1st and 3rd quartiles (25th and 75th percentiles) and whiskers represent the full distribution excluding outliers. Ma, million years ago.

(~90 million years of evolution), with bats resting for 20 h per day and horses for only 3 h (ref. 28).

### Extensive temporal niche partitioning in Tanganyikan cichlids

We next asked whether differences in temporal activity patterns or total amounts of rest are associated with trophic ecology and diet, or morphology. To test this, we correlated our measured activity patterns (PC1 and PC2 loadings, and total rest) with a range of eco-morphological traits and diet guilds of these species[5]. We found that closely related species did not necessarily have similar activity patterns or amounts of total rest (Fig. 2a). No strong phylogenetic signal was observed for diurnal–nocturnal preference (PC1 loadings, Pagel's $\lambda = 0.630$, Blomberg's $K = 0.473$), crepuscularity (PC2 loadings, $\lambda \approx 0$, $K = 0.438$), total rest ($\lambda = 0.609$, $K = 0.525$) or for the multivariate raw speed data ($K_{multi} = 0.411$)[29,30], and behaviours were distributed relatively evenly across tribes (Fig. 2a). For example, strongly nocturnal and diurnal species were observed within the tribes Lamprologini, Ectodini and Tropheini. Crepuscularity was also observed widely and seen in Lamprologini, Limnochromini, Ectodini, Cyprichromini and Trophenini (Fig. 2a). The species with the most rest included *Altolamprologus* spp. (Altshe and Altcom) and *Telmatochromis* spp. (Telshe and Telvit), as well as the mudhole-dwelling *Lamprologus signatus* (Lamsig) (Fig. 2a). Species of the Tropheini had the least rest, and a large diversity in total rest was observed in both Lamprologini and Ectodini (Fig. 2a). These results show that, in Lake Tanganyikan cichlids, both closely related species and highly divergent species have highly divergent activity patterns and total rest amounts.

To test for a correlation between activity patterns and the environment occupied by a species, we compared previously generated stable isotope measurements and datasets of body and jaw morphology[5] to our temporal activity patterns, using a phylogenetically corrected two-block partial least squares analysis (PLS). Stable isotopes measure a species' relative position on the benthic–pelagic axis ($\delta^{13}$C) as well as their relative trophic level ($\delta^{15}$N). These data have previously been linked to morphological adaptations in cichlid body shape and oral and lower pharyngeal jaw morphology, which represent unique adaptations in feeding ecology[5]. In addition, we compared our temporal activity patterns against body size for each cichlid species. Body size is negatively correlated with sleep amounts in herbivorous mammals, with large herbivores spending much of the day grazing for low caloric foods[28]. As has previously been shown across the entire radiation[5], PLS scores for morphological traits strongly and significantly correlated with PLS scores of stable carbon and nitrogen isotope values across the species in our dataset as well (Fig. 2b). PLS scores for body shape were significantly correlated with total rest across the species in our dataset (Fig. 2b,c) and body size was negatively correlated with total rest across cichlid species (Fig. 2d). Specifically, smaller cichlids or those with elongated bodies had the least rest, and larger or deep-bodied cichlids had the most rest (Fig. 2c,d). By contrast, PLS scores representing diurnal, nocturnal and crepuscular preferences of this study were not significantly correlated with either morphology or environment (Fig. 2b and Extended Data Fig. 5b). For example, eye size and shape were not associated with either diurnal/nocturnal (PC1 loadings) or crepuscular preferences (PC2 loadings), in contrast to what has been observed in many other clades[31–34] (Extended Data Fig. 5a). These results reveal that total rest and activity levels, but not temporal activity patterns, are associated with specific morphologies or ecologies in cichlids.

To investigate the associations between temporal activity preferences and total rest with diet guilds, we compared the behaviour of algivores, invertivores, piscivores and zooplanktivores in our dataset. Diurnal–nocturnal preferences, crepuscularity and differences in total rest were evenly distributed across diet guilds, and a phylogenetically corrected analysis of variance (ANOVA) revealed that there are no exclusive relationships between diet guild and activity pattern (Fig. 2e–g and Extended Data Fig. 6). For example, *B. microlepis* (Boumic), *Cyphotilapia frontosa* (Cypfro) and *Lepidiolamprologus elongatus* (Lepelo) are all

medium to large piscivorous cichlids with wide distributions, but have diverse temporal activity patterns, including preferences for diurnal, nocturnal and crepuscular periods, respectively (Fig. 2h). In contrast, the algivore *Tropheus sp.* 'Black' (Tronig), the invertivore *N. buescheri* (Neobue) and the zooplanktivore *Neolamprologus marunguensis* (Neomar) occupied diverse diet niches, but the same diurnal temporal niche (Extended Data Fig. 6). This analysis suggests that cichlids with similar habitat and diet niches can occupy all possible temporal niches, and that cichlids with similar temporal niches can occupy diverse habitat and diet niches.

### Temporal activity patterns in cichlids are part of a continuum bridged by crepuscular states

Our high-resolution and extensive behavioural data allowed us to directly measure the relationship between diurnal–nocturnal preference (PC1 loading), crepuscularity (PC2 loading) and total rest. No significant correlations were observed between total rest and either diurnal–nocturnal preference or crepuscularity, with or without accounting for phylogenetic relatedness (Extended Data Fig. 7). Interestingly, although no linear correlation was observed between diurnal–nocturnal preference (PC1 loadings) and crepuscularity (PC2 loadings), they exhibited a clear parabolic relationship (Fig. 2i). Within our dataset, species with strong diurnal or nocturnal preferences (highly negative or positive PC1 loadings) had weaker crepuscular preferences (low PC2 loadings), and those with strong crepuscular preferences (high PC2 loadings) had weaker diurnal or nocturnal preferences (PC1 loadings near 0). This relationship was confirmed with a quadratic model, with PC1 loadings able to explain between 51% and 60% of variance in PC2 loadings (Fig. 2i). Cathemeral or crepuscular states have been suggested to provide a so-called bridge, facilitating adaptation of physiologies between dramatically different day and night environments[35–37]. The continuum of activity patterns in cichlids that we have observed suggests that an intermediate behavioural state with strong crepuscular preferences might have facilitated shifts between diurnal and nocturnal activity preferences in cichlids.

The tempo of the evolution of morphological traits can also be examined in a phylogenetic context[38], providing evidence for staggered evolutionary bursts in these traits during the adaptive radiation of Tanganyikan cichlids[5]. To further examine the crepuscular bridge hypothesis in a similar phylogenetic context, we performed ancestral state reconstructions for diurnal/nocturnal (PC1 loadings) and crepuscular preference (PC2 loadings) using a Ornstein–Uhlenbeck model (Extended Data Fig. 7d,e; see Methods for more details). As before, we observed a parabolic relationship between PC1 and PC2 loadings when considering both extant and ancestral nodes, and that species only occupy trait space along a bridge-like continuum (Extended Data Fig. 7f). Overlaying the phylogenetic tree structure, we observed that the majority of connections between nodes mirrored the bridge-like continuum. Indeed, when examining the most recent branching events, we observed that ancestral nodes with higher PC2 values resulted in extant species with divergent PC1 values (for example, *N. brichardi* and *Neolamprologus helianthus* or *L. elongatus* and *Lepidiolamprologus attenuatus*). This is in contrast to ancestral nodes with lower PC2 values, which branch into extant species with similar PC1 values (for example, *T. moorii* and *Tropheus niger* or *N. toae* and *Variabilichromis moorii*) (Extended Data Fig. 7g). Given that temporal activity patterns can also be categorized into discrete bins ('diurnal', 'nocturnal', 'crepuscular' or 'cathemeral'), we also used evolutionary models for discrete characters. We compared the performance of models that allowed direct transitions between diurnal and nocturnal states (equal-rates (ER), symmetrical rates (SYM) and all-rates different (ARD) models) to constrained models where direct transitions between diurnal and nocturnal states were not allowed (bridge-only-SYM, bridge-only-ARD models), allowing us to directly test the crepuscular bridge hypothesis (Extended Data Fig. 7h). Comparison of Akaike information criterion (AIC) scores favoured acceptance of the ER and bridge-only-SYM

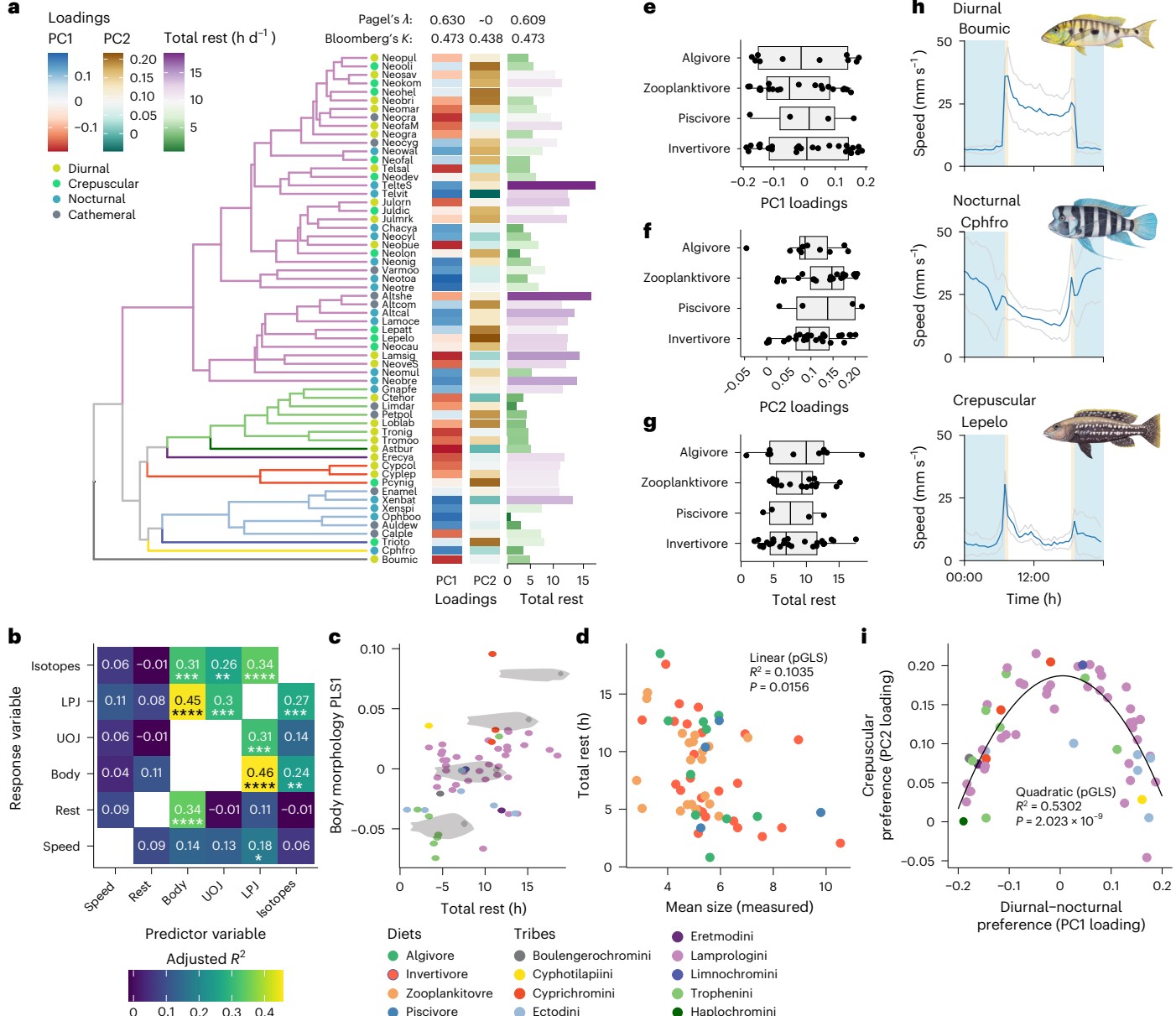

**Fig. 2 | The weak relationships between behaviour and ecological measures support temporal niche partitioning. a**, The phylogenetic tree of the species in our study along with diurnal–nocturnal preference (PC1 loadings), crepuscular preference (PC2 loadings), temporal activity pattern and total rest, along with values for Pagel's $\lambda$ and Bloomberg's $K$ for each trait. **b**, Heatmap of adjusted $R^2$ values for pairwise phylogenetically corrected two-block PLS analysis for comparisons between PC1 loadings, PC2 loadings, total rest and published data for stable isotopes values and datasets of body and jaw morphology[5]. Numbers and colour represent $R^2$ values; *$P < 0.05$, **$P < 0.01$, ***$P < 0.001$. **c**, Scatterplot of total rest for each species and values for the PLS of body morphology that most correlates with it. Dots are coloured by tribe and silhouettes indicate morphologies along the trend line (lowest rest to highest rest). **d–g**, Scatterplot of total rest for each species and the mean size across individuals in our experiment (**d**). Dots are coloured by diet guild. Inset displays the statistics from the fitting of a linear regression using pGLS. Diet guilds plotted against PC1 loadings (**e**), PC2 loadings (**f**) and total rest (**g**); each dot is one species. The centre line of each box represents the median, the box represents the 1st and 3rd quartiles (25th and 75th percentiles) and whiskers represent the full distribution excluding outliers. **h**, Examples of three species from the piscivore diet guild with diverse daily speed patterns (speed mean +/- s.d.). Species names are abbreviated using a six-letter code (Supplementary Data 1). **i**, The relationship between PC1 loadings and PC2 loadings can be best explained by a quadratic function. Species names are abbreviated using a six-letter code following ref. 5 (Supplementary Data 1). Inset displays the statistics from the fitting of a linear regression using pGLS. LPJ, lower pharyngeal jaw; UOJ, upper oral jaw.

models over unconstrained models (Extended Data Fig. 7h). Under the bridge-only-SYM model, high transition rates were observed between nocturnal–crepuscular, nocturnal–cathemeral and diurnal–crepuscular states, with no transitions between diurnal–nocturnal states and between cathemeral–diurnal states. Together these results suggest that cichlids have transitioned between diurnal and nocturnal states through an intermediary crepuscular bridge.

## Diel activity patterns are associated with synapse function and genes associated with neurological disorders, not the circadian clock

To investigate the genetic basis of temporal activity patterns and total rest, we followed an approach used in a recent study on the genetic basis of exploratory behaviour in Lake Tanganyika cichlids[39] (Supplementary Information). This method uses a modified genome-wide association

study (GWAS) approach to identify highly associated variants (HAVs) with each trait (Fig. 3a,b, Extended Data Fig. 8 and Supplementary Information). Possible effects or associations between HAVs and nearby genes were identified using SnpEff, and we termed all genes that were annotated by SnpEff as potential highly associated genes (HAGs) (Supplementary Data 3)[40,41]. This approach identified 848 HAGs with diurnal–nocturnal preference, 850 HAGs for crepuscularity and 793 HAGs for total rest. Unlike with HAVs themselves, we did observe an enrichment in the pairwise overlaps between HAGs for all three behaviours (Fig. 3c). The top HAVs for temporal activity patterns and total rest were associated with genes with known functions in the nervous system, including the regulation of sleep, or genes associated with neuronal disease and dysfunction in humans and model organisms.

For example, the most significantly associated variant to diurnal–nocturnal preference identified by phylogenetically generalized least squares (pGLS)-GWAS was within the intron of the gene *tpst1l* and downstream of both *tmtops3b* and *crcp* (Fig. 3d and Extended Data Fig. 9a). *Teleost multiple tissue opsin 3b* (*tmtops3b*) is a non-visual opsin that may allow detection of blue light in hypothalamic deep brain nuclei[42]. Though not the closest gene to the HAV, *crcp* is the receptor for a neuropeptide (CGRP) with functions similar to hypocretin/orexin and PDF in regulating rest/wake in zebrafish and flies, respectively, and acts downstream of circadian pacemaking neurons[43,44] (Fig. 3d). A single nucleotide polymorphism (SNP) in the intron of *cacnb3b* was also strongly associated with diurnal–nocturnal preference. While *cacnb3b* is a subunit of the voltage gated calcium channel, another subunit of this channel (CACNA1C) has been linked to sleep latency and quality across multiple human populations[45,46].

Two of the top HAVs for crepuscularity were a SNP in an intron of the pyruvate carboxylase gene (*pc*) and one downstream of the *nsg2* gene (Fig. 3e and Extended Data Fig. 9b). Pyruvate carboxylase is involved in the production of neurotransmitters, and its deficiency is associated with both neurodevelopmental defects and seizures/epilepsy[47]. The *nsg2* gene is required for normal synapse maturation and regulates excitatory neurotransmission[48]. Mice lacking *nsg2* also display reduced activity at night, suggesting it is involved in temporal activity preferences as well[49]. For total rest, our analysis also identified two adjacent SNPs separated by one base pair in the fifth to last intron of the carbamoyl-phosphate synthase 1 gene (*cps1*) and one SNP in the fourth intron of the gene encoding double C2-like domains beta (*doc2b*) (Fig. 3f and Extended Data Fig. 9c). While *cps1* has been linked to diseases associated with lethargy[50], *doc2b* regulates spontaneous neurotransmitter release in the hippocampus[51,52].

To more systematically interrogate the functions of genes associated with temporal activity patterns, we performed gene ontology (GO) enrichment of HAVs using SNP2GO[41]. This approach tests for the overrepresentation of SNPs nearby to genes associated with specific ontologies, controlling for gene size and other local genomic effects. This analysis implicated pathways and GO terms associated with synaptic transmission or assembly, and human neurological disorders, including attention deficit disorder (ADHD), epilepsy, and depression associated with temporal activity patterns and total rest across cichlids (Fig. 3g,h). Despite the importance of light and the internal circadian clock for cichlid activity patterns[14] (Extended Data Fig. 10a–d), functions related to the regulation of the circadian clock, or melatonin regulation/signalling were not observed to be enriched in the HAVs (Fig. 3c–f and Extended Data Figs. 9 and 10e–g). Together, these results suggest that evolutionary transitions between activity patterns, including diurnal to nocturnal transitions, are associated with genes regulating synaptic neurotransmission, rather than those involved in the circadian clock.

## Discussion

Here, using the largest study of its kind, we characterized the temporal activity patterns of 60 ecologically diverse species of cichlid fishes from Lake Tanganyika. We show that these cichlid species can display a wide range of temporal activity patterns, which may have contributed towards the diversification of these species through temporal niche partitioning. Our work extends on a study examining the activity patterns of 11 cichlid species over a 24-h period from the Lake Malawi cichlid adaptive radiation[14]. In that study, while most species were diurnal or had no clear rhythm, one displayed nocturnal activity, prompting the authors to speculate that different activity patterns may have contributed to niche partitioning within that lake[14]. Our study provides extensive evidence of temporal niche partitioning by examining in-depth (6 days and nights) a large number of species (*n* = 60) from a well-characterized adaptive radiation in a 'common garden' setup. Moreover, by integrating behavioural data with eco-morphological data for each species, we observe that cichlid species which occupy similar habitat and diet niches feature different temporal activity niches, and species with similar temporal niches differ in their habitat, morphology and diet specializations. These results demonstrate that this diversity of temporal patterns is an independent axis of diversification.

Temporal niche partitioning has mainly been observed in distantly related species[7,9,53]. For example, temporal niche partitioning was observed in large coastal sharks[8] and in mammals[9,54]. Conversely, several studies have looked at the temporal activity patterns of closely related, but non-sympatric *Drosophila* species, and have found that these species all have similar temporal patterns, but differ most in their amount of activity[55,56]. Detailed studies of more species and other adaptive radiations in ecologically relevant contexts (within social groups such as in Lloyd et al., 2023 (ref. 57) and more complex environments) as well as studies in the wild will be required to understand the full complexity of daily activity patterns.

Furthermore, our work suggests that transitions between diurnal and nocturnal activity patterns were probably facilitated through an intermediate crepuscular state or 'bridge'. This observation confirms and extends previous categorical studies on the temporal activity patterns of skinks[58], geckos[59] and mammals[35–37,60], in which direct evolutionary transitions between diurnal and nocturnal activity patterns are slower than transitions from crepuscular/cathemeral to diurnal and nocturnal patterns. Our study provides high-resolution quantitative, rather than categorical, behavioural evidence, as well as ancestral reconstructions and discrete character evolutionary modelling. We observe that species occupy a continuum of behavioural states consistent with this bridge hypothesis, and that transitions between diurnal and nocturnal activity patterns occur through a crepuscular intermediary state. Though these analyses are consistent with a crepuscular bridge hypothesis, our dataset has limited taxonomic coverage. Reconstructions of temporal activity across a more complete cichlid phylogeny, including monophyletic sister species and representation from all Tanganyikan cichlid tribes, could provide stronger evidence of the 'bridge' state at ancestral nodes where transitions are predicted to have occurred, and determine the tempo and dynamics of temporal niche partitioning in this system. This would also allow us to test whether temporal niche partitioning evolved after, or drove, diversification, similarly to that seen for morphological traits[5].

Our study also provides an important set of candidate loci that may underlie specific temporal activity patterns. Temporal activity patterns have a polygenic basis in cichlids, as evidenced by many SNPs across the genomes being associated with each trait. It has been suggested that polygenic trait architectures are better than simple architectures at promoting rapid and stable speciation in sympatry[61,62]. Interestingly, we found no evidence for the involvement of circadian clock genes in diurnal, nocturnal and crepuscular preferences, or in total rest. This lack of associations with clock genes suggests that evolutionary transitions in activity patterns occur independently of, and downstream of, the core circadian clock. This could be a reflection of the plasticity of the circadian clock system, with diurnal and nocturnal mammals having genetically and functionally conserved circadian timing systems[63–66].

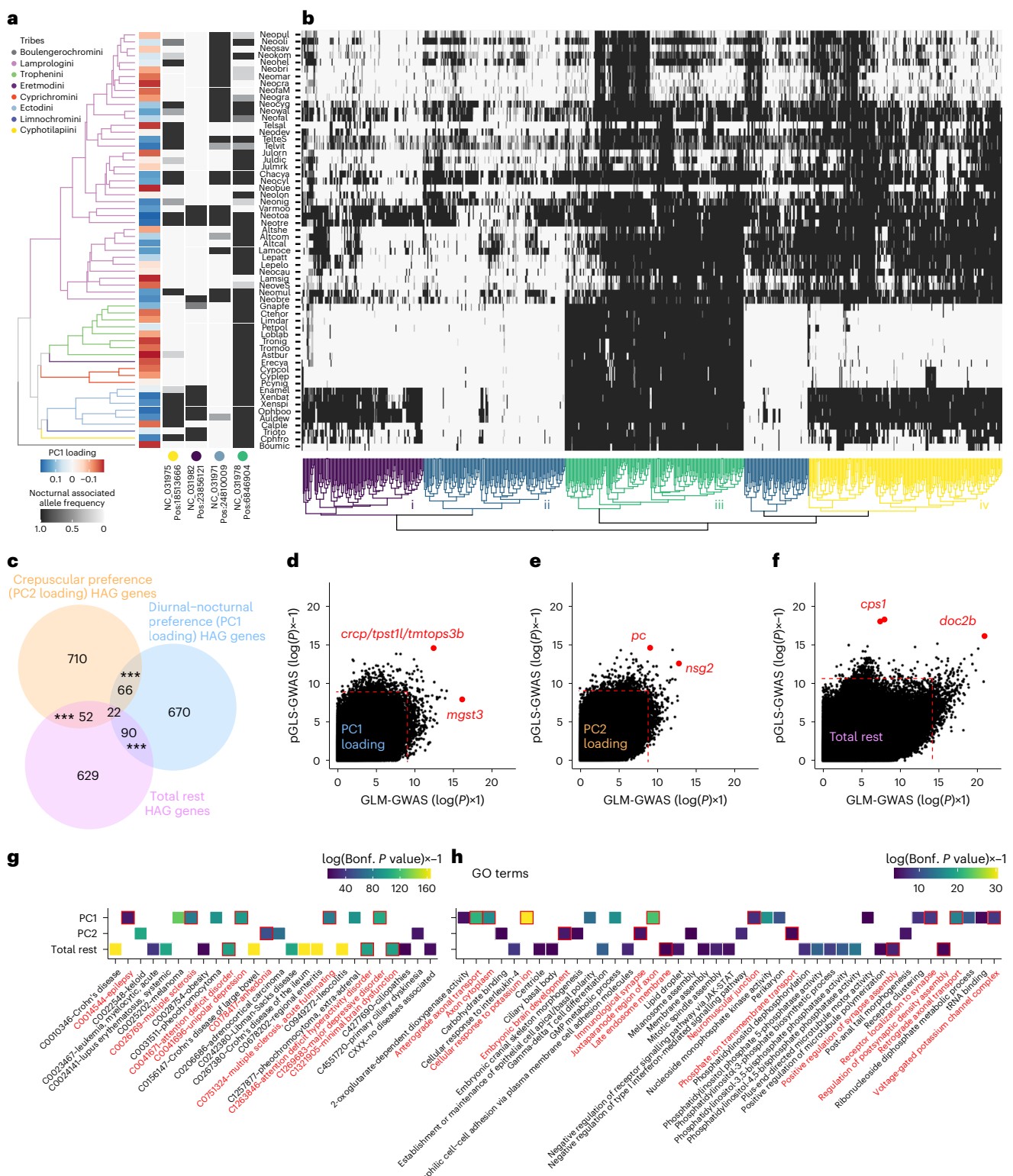

**Fig. 3 | Genome-wide signatures of cichlid temporal activity patterns.**
**a,b**, Phylogenetic tree of the species in our dataset along with the PC1 loadings
(diurnal–nocturnal preference) and the allele frequencies of the high PC1 loading
associated allele at four representative HAV loci (**a**). The coloured dot represents
the cluster that each HAV is representative of from **b. b**, Heatmap of the allele
frequencies of the high PC1 loading associated allele across all HAVs for PC1 loading
(diurnal–nocturnal preference). Colours on the dendrogram represent groups
of alleles with similar patterns across species (i, ii, iii, iv and v; see Supplementary
Information). **c**, Venn diagram of the overlap of HAGs for PC1 loadings, PC2 loadings
and for total rest, annotated with the number of genes in each category. ***$P < 0.001$

for enrichment tests between pairwise gene sets. **d–f**, Scatterplots of the pGLS-
GWAS $P$ value and GLM-GWAS $P$ values for all SNPs associated with PC1 loadings (**d**),
PC2 loadings (**e**) and total rest (**f**). The SNPs with the lowest $P$ value in the pGLS and
GLM tests are labelled. Statistics are derived from the fitting of a linear regression
(GLM-GWAS) or linear regression using pGLS (pGLS-GWAS). Dotted lines indicated
genome-wide cutoffs for identification of HAVs. **g,h**, Tile-plots of human diseases (**g**)
and GO terms (**h**) identified by GO analysis with SNP2GO showing enrichment for
the human orthologs of genes nearby to HAVs for PC1 loadings, PC2 loadings and total
rest. Tiles are coloured by the Bonferroni corrected $P$ value for the association test.
GO categories associated with neuronal and synaptic function are highlighted in red.

For example, in some species of rodents, while the central clock (the suprachiasmatic nucleus) maintains its phase between both diurnal and nocturnal species, many non-suprachiasmatic-nucleus brain regions can be anti-phase between diurnal and nocturnal animals, and in phase with their activity rhythms[67–69]. It is unclear if teleost fishes have a central clock, and many of their tissues are directly light-responsive, suggesting their tissue-specific clocks may be similarly uncoupled. Perhaps these factors could allow for even more circadian pattern flexibility depending on interpretation of the zeitgebers ('time givers', environmental cues that can entrain the circadian rhythm) each tissue uses to entrain their intrinsic clocks[70]. Indeed, modelling studies have also demonstrated that changes in the activity phase of circuits downstream of the central clock can explain switches between diurnal and nocturnal behaviour[71]. Moreover, environmental signals known as masking factors can change the output of internal clocks without necessarily changing the underlying clock rhythm[72,73]. For example, in the presence of predatory foxes, nocturnal Norway rats switch from nocturnal to diurnal activity patterns[74]. Given the fast divergence of temporal activity patterns in cichlids, it would be interesting to test these species under different conditions to see if environmental masking factors or zeitgebers could affect their activity patterns.

In our GWAS analysis, we observed enrichment for genes associated with neuronal function and synaptic transmission associated with differences in temporal activity patterns. Many of these genes are also associated with human neurological and neuropsychiatric disorders, including Alzheimer's, ADHD and epilepsy/seizures, as well as social behaviours. Furthermore, sleep phenotypes and neurological disorders have overlapping genetic components in humans[75], and neuropeptides controlling sleep and wake states have been linked to the occurrence and severity of seizures[76]. Our results also suggest an overlap between the molecular mechanisms underlying evolutionary transitions in activity patterns and sleep duration and quality in humans from GWAS studies[45,46]. However, we cannot rule out that some of these associated SNPs could be due to the specific environmental conditions of the behavioural setup rather than to the activity timing preferences. For example, cichlid species have varying levels of sociality and aggression across species, and our measured activity patterns could be affected by species-specific responses to the semi-isolation of our setup. Future work and functional experiments (allele-swaps, gene knockdowns or knockouts) will be necessary to determine which SNPs and genes are causally linked to evolutionary shifts in activity patterns in cichlids, and whether these are the same mechanisms governing shifts across other clades, including mammals.

## Methods

### Fish husbandry

All animal work was performed at the facilities of the University of Basel. The initial screen of 60 cichlid species was performed at the Zoological Institute, University of Basel. All species in our study were housed in similar conditions (12 h of light, 12 h of darkness; 24–25 °C water temperature) with 30-min ramping light conditions at dawn and dusk. Cichlids were fed every couple of days at inconsistent times during the day with various foods to best suit the ecology of the species, with fortnightly water changes. The exception was the dark:dark experiments that were performed at the Biozentrum (University of Basel). Here, species were housed in a recirculating system (Tecniplast) with 8% exchange of water every day and 14 h of light, 10 h of darkness, 26 °C water temperature and 15-min ramping light conditions at dawn and dusk. Fish were fed twice a day, once with live food and once with dry food.

Cichlids were kept at densities no greater than 0.5 cm of fish body length per litre of water for fish up to 5 cm total length and 1 cm of fish body length per litre of water for larger fish up to 15 cm total length. They were generally kept in species-specific tanks. Environmental enrichment varied between species according to their ecology, but generally consisted of rocks and terracotta plant pots serving as refuge and breeding areas, sandy or gravel substrates and shells. The source of animals varied, but they stem generally from in-house natural breeding of populations originally collected in Lake Tanganyika or were purchased from local suppliers which source fishes from both wild and aquarium populations (Garten- und Zoobedarf Schrepfer). All experiments were performed under holding permit nos. 1010H and 1035H and experimental permit nos. 2356 and 3102 issued by the cantonal veterinary office in Basel.

### Behavioural assays

We developed a behavioural assay based on a previous study[19] to record daily activity patterns of individual fish. Glass tanks (45 cm height × 110 cm length × 25 cm depth, clear glass on one long side and opaque glass on three sides, Pavlica Akvária) were used to house fish for experiments. Each tank had a thin layer of sand and was physically divided into arenas for individual fish. For most species the tanks were divided into four arenas (each 25 cm wide), though for the larger fish they were divided into larger arenas. For *B. microlepis* (Boumic) juveniles, tanks were split into three arenas (33 cm wide). For *E. melanogenys* (Enamel) and the larger individuals of *L. dardennii* (Limdar), tanks were split into two arenas (50 cm wide). The dividers were made of PMMA opal white with a mesh insert allowing for water flow and visual communication of fish to their neighbours. The feeding schedule was consistent with that of the home tanks, which allowed the effects of feeding to be averaged out over the six recording days. Fish were kept on a 12:12 h light:dark cycle from 07:00 to 19:00 CET with the light ramping on from 07:00–07.30 and ramping down from 18.30–19:00 (TC420 light controller), with the exception of the species used in the dark:dark experiments, which were housed on a 14:10 h light:dark cycle from 08:00 to 22:00 with the light ramping on from 08:00–08.30 and ramping down from 21:30–22:00. This was to account for the fish being kept in a different aquarium room with a 14:10 h cycle instead of the 12:12 h cycle. The tanks were backlit with a panel of infrared LED lights which were diffused by the opaque glass and a diffuser to generate even lighting and allow recording during the night.

Using this setup we assayed 60 species of cichlid fishes from Lake Tanganyika (Fig. 1a). We aimed to assay 7 to 12 fish per species, but in some cases we included rarer species for which we only had fewer than 7 individuals (*Chalinochromis cyanophleps* (Chacya), *Lobochilotes labiatus* (Loblab): 2 individuals; *Altolamprologus compressiceps* (Altcom): 3 individuals; *Neolamprologus walteri* (Neowal): 4 individuals; *Eretmodus cyanostictus* (Eracya), *Neolamprologus falcicula* (Neofal), *Neolamprologus sp. 'Kombe'* (Neokom), *Xenotilapia spilopterus* (Xenspi): 6 individuals; Supplementary Data 1). The six-letter species code follows Ronco et al.[5] except for *Telmatochromis* sp. "Lufubu" (Telluf) which has been renamed *Telmatochromis salzburgeri* (Telsal)[27]. Because the fish and the behavioural setup were kept in a room where the temperature could not be well controlled, temperature varied depending on season and was generally between 24 °C and 28 °C. Animals that became sick, died or escaped were excluded. Age of tested fish varied; almost all fish were tested when they had indicators of sexual maturity (colouration, behaviours, breeding), although *B. microlepis* (Boumic) and *C. frontosa* (Cypfro) were tested as juveniles due to the long life cycles and lack of availability of adults of these larger species. Both male and female fishes were tested, although we did not have enough individuals of each sex to systematically compare across species. The daily activity measures for both sexes are plotted in Extended Data Fig. 2.

As part of a pilot study we recorded several species from our study in their home tanks. These home tanks are environmentally enriched and the animals had formed stable social groups, but precluded high-throughput behavioural tracking and analysis. To test if species had similar activity patterns in both home tanks and our reductionist assay, we selected three species which had different activity patterns in the high-throughput assay to quantify in the home tank recordings

(Extended Data Fig. 3). We manually quantified how many fish were visible and active every 3 min across roughly 16 h of video per species. Non-visible fish were assumed to be motionless in their hiding places. The data was binned into 30-min increments for plotting.

## Processing of behavioural data

Each arena was recorded from the front using cameras (RoHS 1.3MP B&W Chameleon USB 3.0 Camera 1/3″ CCD CS-Mount CM3-U3-13S2M-CS, Flir) fitted with lenses (YV4.3 × 2.8SA-2, Fujinon) and long-pass filters to exclude white light (MidOpt Near-IR Longpass slipmount Filter 780-30.5). Recording was done by custom Python3 v.3.7 code using the Spinnaker SDK API, with processing from the following packages: NumPy[77], imageio[78], opencv-python[79], PyYAML, pandas[80] and Matplotlib[81]. Each arena was selected as a region of interest (ROI), and a 10 fps mp4 video of this ROI was saved every hour. We tracked animals using background subtraction and thresholding with a minimum object area of 100 pixels. Backgrounds were calculated off the 95th percentile of pixel intensity from the previous 1 h video. Positions of fish were interpolated during frames where the fish was not visible (for example, if it was hidden in a corner or behind sand). Timestamps, $X$ and $Y$ position and surface area of the tracked object were saved alongside each video as a comma-separated file (csv), as well as a YAML file of metadata, using PyYAML v.5.3.1 software[82]. Fish were recorded for a week, but only data from midnight on the first day to midnight on the second-last day were used for further analysis (six days/nights total).

Extensive quality control was performed on each video to ensure high accuracy of behavioural tracks across cichlid species. This primarily involved excluding portions where the tanks were briefly obstructed by personnel, or decreasing background generation periods to 30-min, 20-min or 10-min periods (instead of 1 h) to account for sand displacement by individual fish—when a fish moved a large amount of sand within an hour, this could lead to the sand being tracked instead of the fish. These shorter periods for background calculation meant less difference in the sand between the tracked frame and the background, allowing for more accurate tracking of the fish. Additionally, some videos were retracked with smaller ROIs to remove problematic regions associated with the edges of the original ROIs chosen. This included removal of pixels covering adjacent arenas (which can permit detection of adjacent fish) or pixels covering the top of the tank, which could contain disturbances in the water due to feeding or water exchange (water bubbles).

Swimming speed was calculated using subsequent $X$ and $Y$ positions. Video frames where speed exceeded a threshold of 200 pixels were replaced with the average of ±5 frames (1 s). Such high speeds were never observed for individual fish, but represented jumps caused by the background subtraction detecting objects elsewhere in the tank (water bubbles or movement in adjacent arenas). Speed as well as $X$ and $Y$ position were smoothed by 0.5-s windows. For most plots data is binned by 30 min (Supplementary Data 2).

## Exploration of behavioural data

To investigate the patterns of daily activity we used PCA. Averages of the daily speed per species in 30-min bins were standardized by $z$-scoring (48 dimensions). PCA was run with ten components using the PCA function from the sklearn.decomposition function from the scikit-learn v.0.24.0 package[83]. Ward clustering of PC1 and PC2 loadings separated the species into three groups, which, based on the individual species patterns, we defined as diurnal, nocturnal and crepuscular (Fig. 1g). However, as we ran PCA on the daily averages, this analysis does not take individual variability into account, and we saw species with variable patterns across and between individuals (Extended Data Fig. 1). We therefore designated any species where the minimum to maximum daily average speed difference was smaller than two mean standard deviations to be cathemeral (see Fig. 1g).

Besides speed, we also transformed activity into a measure of rest. First, movement was determined by a threshold of 15 mm s⁻¹; above this threshold the animal is actively moving, while below this threshold most activity is noise (artefacts of tracking or small undulations due to fin movements). To uncover behavioural states where the fish was mostly inactive, we used *rest*. A time point was set to be positive for rest when there was less than 5% movement in a sliding 60-s window. Like speed, rest was also binned in 30-min bins.

We also quantified the position of fish. Here we scaled the data between 0 and 1, where 0 was the minimum and 1 was the maximum fish position. This allowed for the absolute top position to be 1 and the absolute bottom position of the fish to be 0. This was necessary as we tracked the centroid and, because different fish had different sizes, a smaller, thinner fish might look like it was closer to the bottom compared to a bigger, taller fish. Paired $t$-tests with Bonferroni correction were used to test for significant differences for vertical rest position between active (non-rest) and rest states.

To find rhythmicity in the speed data we used periodograms. We calculated these using code from the CosinorPy package v.3.0 using the 30-min binned speed data[84]. The default CosinorPy threshold (0.05) was used to identify significant spectra peaks.

Plots were made using the software packages Matplotlib[81], Seaborn[85] and CMasher[86].

## Comparing behaviour to ecological features

Eco-morphological data for all cichlid species in our study were taken from Ronco et al.[5]. Three of our focal species are not included in the Ronco et al. dataset as they are either not part of the radiation or are not found within the lake. These excluded species are: *A. burtoni* (Astbur), *Neolamprologus devosi* (Neodev) and *T. salzburgeri* (Telsal). We used data on body, upper oral jaw and lower pharyngeal jaw morphology, and stable isotope values ($\delta^{13}C$, $\delta^{15}N$). We compared these values to the metrics derived above, specifically diurnal–nocturnal preference (PC1 loadings), crepuscular preference (PC2 loadings) and total rest per species.

Phylogenetic signal in temporal activity pattern traits (PC1 and PC2 loadings, total rest) was tested using the function phylosig from the package phytools[87] and the function physignal from the package geomorph[88] for testing the multivariate speed data. To investigate the links between eco-morphological traits and temporal activity patterns we performed pairwise phylogenetically corrected two-block PLS, alternating each trait between predictor and response variables using the functions two.b.pls and pls from the R packages pls[89] and geomorph[88,90]. Importantly, PC1 and PC2 for temporal activity patterns which were calculated by PLS represented day–night preference and crepuscularity, and were highly similar to principal components for activity patterns (PC1 and PC2 in Fig. 1). To investigate links between cichlid diet guilds and temporal activity patterns, cichlids were grouped by diet and a phylogenetic ANOVA was performed using the function aov.phylo from the geiger package[91].

## Ancestral reconstructions of activity patterns

We performed phylogenetic analysis and ancestral reconstructions in R using the packages geiger[91] and phytools[87]. For continuous variables, we compared brownian motion (BM), Ornstein–Uhlenbeck (OU) and early-burst (EB) models using the geiger function fitContinuous. Model fits were compared using corrected AIC scores; for PC1 loading (BM = −68.3, OU = −72.8, EB = −66), for PC2 loadings (BM = −150.2, OU = −160.1, EB = −147), which support rejection of BM and EB models in favour of an OU model. To reconstruct ancestral states under an OU model, we used the function anc.ML from the phytools package. Reconstructions using BM or EB models result in highly similar reconstructions, including for those specific branch points identified in Extended Data Fig. 7g. For discrete characters, we used the function fitMK from the phytools package to compare the performance of ER, SYM and ARD

models, as well as the performance of constrained versions of the SYM and ARD models where direct transitions between diurnal and nocturnal states were not allowed (bridge-only-SYM, bridge-only-ARD). These constrained models mimic the crepuscular/cathemeral bridge hypothesis, allowing us to test the hypothesis against models where direct transitions are allowed. We compared AIC scores between models, which account for the likelihood of the model, while penalizing models with a high number of parameters (to avoid the effect of overfitting) (Extended Data Fig. 7h).

### Identifying genetic variants associated with temporal activity patterns

Whole genome sequences were obtained from GenBank; the accession numbers for the samples used are available in Supplementary Data 4 (ref. 5). This dataset contains two individuals (one female and one male) from each species included in our behavioural analysis. Therefore, whole genome sequences from 60 species were used to generate a new variant call set specific to this study based on alignment to the *Oreochromis niloticus* genome (O_niloticus_UMD_NMBU, GCF_001858045.2, NCBI). We followed Genome Analysis Toolkit (GATK) best practices and analysis pipelines to align and call variants (GATK v.4.2.4.0) and used genome masks generated with custom scripts for variant filtration. This was followed by association studies to identify HAVs, and analysis of genes nearby to HAVs with custom R scripts. All steps are outlined in more detail in the following sections.

### Alignment, variant calling and variant filtration

Short reads from each individual fish were processed using MarkIlluminaAdaptors before being aligned to the most recent *O. niloticus* genome assembly (O_niloticus_UMD_NMBU, GCF_001858045.2, NCBI) using bwa mem software (v.0.7.17)[92]. Aligned reads were processed using MergeBamAlignment, SortSam and MarkDuplicates in Picard (v.2.26.2)[93]. SNPs and short insertions and deletions were detected against the reference genome for each individual using HaplotypeCaller across 80 genomic intervals of equal length. Variant calls across genomic intervals were then combined using GatherVcfs and the resulting variant call format (vcf) files for each individual were collected using CombineGVCFs. All samples were then jointly genotyped using GenotypeGVCFs, resulting in a single vcf file containing variant calls across all sites and for all 60 species.

We performed extensive filtering on the genotyped variant file using three separate genome masks. The first genome mask was generated by identifying low-quality sites using the VariantFiltration function and the expression "QD < 2.0 || FS > 60.0 || MQ < 40.0 || MQRankSum < −12.5 || ReadPosRankSum < −8.0" in GATK. The second genome mask identified variants whose read depth was not less than 900 or greater than 1,900 when summed across all samples. Cutoffs for read depth were determined by examining the distribution of read depth from a random subset (10%) of genotyped sites and identifying a region with roughly normal distribution. The third genome mask was based on the ability of pseudo-reads generated from the *O. niloticus* genome to reliably map back to the correct location. We used the SNPable tool to divide the reference genome into overlapping 100 *k*-mer sequences (http://lh3lh3.users.sourceforge.net/snpable.shtml) and to generate an intermediate mask. This mask was then converted into bed format using a modified version of the makeMappabilityMask Python script from msmctools (https://github.com/stschiff/msmc-tools/tree/master). All three masks were then merged and used to hard filter genotype variants using VariantFiltration and SelectVariants.

This list of variants that passed the above masking approach was subjected to one final filtering step to simplify association studies. We excluded sites where the minor allele was present in only a single individual, all multiallelic sites, sites with insertions or deletions, and those that mapped to unplaced scaffolds. This filtering pipeline resulted in the identification and selection of roughly 39 million SNPs.

### Genome-wide association of variants to temporal activity patterns

To associate variants with temporal activity pattern preferences and total rest, we first estimated allele frequencies for each SNP using the evo software package v.0.1r23 and the subprogram alleleFreq following the approach of Sommer-Trembo et al.[39] (https://github.com/millanek/evo). Briefly, we used genotype likelihoods from GATK and assumed a Hardy–Weinberg prior to obtain posterior probabilities for reference and alternative allele frequencies at each loci. Allele frequencies derived using this approach were then used to test for associations between temporal activity pattern preferences and total rest.

We used a combination of a general linear modelling (GLM) and pGLS to identify associated variants, which accounts for phylogenetic relationships, as well as the possibility of allele-sharing between species. In association tests, diurnal–nocturnal preference (PC1 loadings), crepuscular preference (PC2) and total rest were used as response variables, and linearly scaled allele frequencies (ranging between −1 and 1) at each of the 40 million SNPs as the predictor. Both models were run in the R environment (v.4.0.3). The GLM was run using the command lm(temporal activity phenotype ~ allele frequency) and iterated over each SNP. For pGLS we used the caper package[94] and the function pGLS using the command pGLS(temporal activity phenotype ~ allele frequency, phylogenetic tree), where the phylogenetic tree was that of the Lake Tanganyika cichlids from ref. 5 pruned to include only the focal species in our study.

### Reporting summary

Further information on research design is available in the Nature Portfolio Reporting Summary linked to this article.

### Data availability

The time-calibrated species tree, morphology and stable carbon and nitrogen isotope signatures were taken from Ronco et al.[5] (data available via Dryad at https://datadryad.org/stash/dataset/doi:10.5061/dryad.9w0vt4bbf). The binned behavioural data used for the figures are available in Supplementary Data 1–4; the unbinned behavioural tracks are available via Dryad at https://datadryad.org/dataset/doi:10.5061/dryad.j0zpc86sv (ref. 95). Source data are provided with this paper.

### Code availability

Scripts for recording and tracking were written in Python and are available online[96] (https://github.com/annnic/cichlid-tracking). All scripts for performing quality control and running analysis of behavioural activity, including generation of plots of cichlid weekly and daily speeds, were written in Python and are available online[97] (https://github.com/annnic/cichlid-analysis). Scripts for analysis of eco-morphological data, construction of phylogenetic plots, HAV analysis and GO analysis were written in R and are available online[98] (https://github.com/maxshafer/cichlid_sleep_gwas). Scripts for running GWAS, including the GATK Python, generation of genome masks and variant identification and filtering, were written in Bash and are available online[98] (https://github.com/maxshafer/cichlid_sleep_gwas).

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

## Acknowledgements

We thank members of the Shafer, Schier and Salzburger laboratories for discussion and advice and A. Kempf and V. Bitsikas for valuable comments on this manuscript. We thank the Biozentrum mechanical and electrical workshops as well as D. Lüscher for technical support, J. Johnson for fish illustrations and K. Ntemos at the University of Basel, sciCORE for discussions. This work was supported by grants from the National Sciences and Engineering Research Council of Canada to M.E.R.S. (no. RGPIN-2024-05509), the Swiss National Science Foundation to M.E.R.S. (no. 196313), A.F.S. (no. 197837), M.M. (no. 193464) and W.S. (no. 208002), from the Human Frontier Science Program to A.L.A.N (no. LT000400/2019-L) and by the German Science Foundation (fellowship SO 1737/1-1) and the Research Fund for Junior Researchers of the University of Basel to C.S.-T. Computing resources and infrastructure were provided by sciCORE (https://scicore.unibas.ch/), the Center for Scientific Computing at the University of Basel, with support from the Swiss Institute of Bioinformatics.

## Author contributions

A.L.A.N. and M.E.R.S. contributed equally as first authors; A.M. and A.A.-W. contributed equally as co-authors. A.L.A.N., M.E.R.S., W.S. and A.F.S. conceived and designed the study. A.L.A.N. and M.E.R.S. collected and analysed behavioural data, performed ecological comparisons and carried out the genomic analysis. A.I., A.R., R.G.-D. and L.F. aided in study design and collection and interpretation of behavioural data. M.M. and C.S.-T. provided scripts and aided genomic analysis. A.M. and A.A.-W. quantified behavioural data and aided in interpretation of results. A.L.A.N., M.E.R.S., W.S. and A.F.S. interpreted the results and wrote the manuscript. All authors read and approved the manuscript.

## Competing interests

The authors declare no competing interests.

## Additional information

**Extended data** is available for this paper at https://doi.org/10.1038/s41559-025-02819-z.

**Correspondence and requests for materials** should be addressed to Maxwell E. R. Shafer.

[1]Biozentrum, University of Basel, Basel, Switzerland. [2]Zoological Institute, Department of Environmental Sciences, University of Basel, Basel, Switzerland. [3]Department of Biology, Institute of Ecology and Evolution, University of Bern, Bern, Switzerland. [4]Department of Paleontology, University of Zurich, Zurich, Switzerland. [5]Present address: Department of Cell and Systems Biology, University of Toronto, Toronto, Ontario, Canada. [6]These authors contributed equally: Annika L. A. Nichols, Maxwell E. R. Shafer. ✉e-mail: maxwell.shafer@utoronto.ca

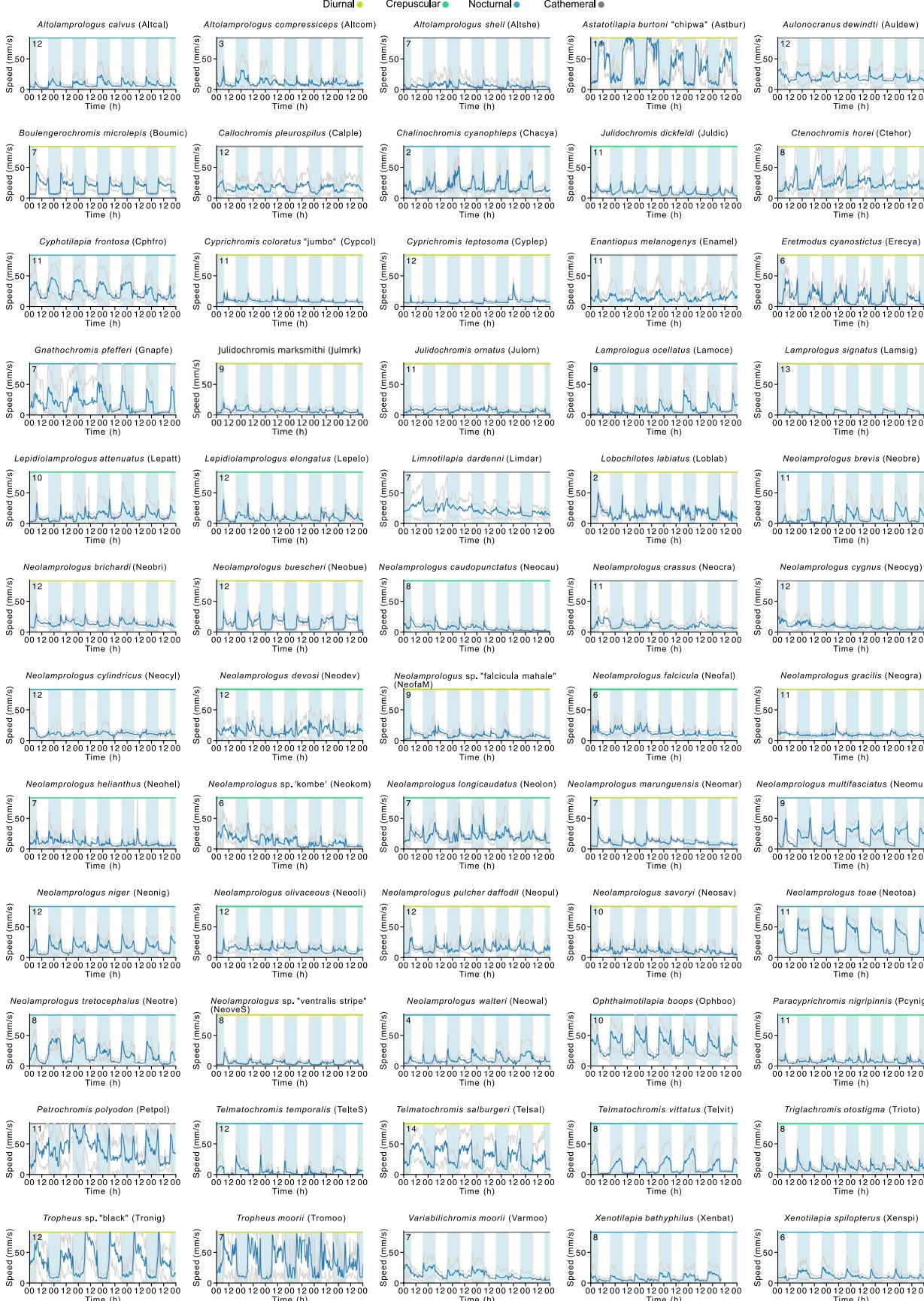

**Extended Data Fig. 1 | Weekly speed for cichlid species in this study. a**, Weekly speed (mean +/- SD) traces for the 60 cichlid species assayed. Timeline and light cycle (12:12 h light:dark) is the same as in the schematic shown in Fig. 1b. Number of animals assayed shown in the top left corner. Temporal guild shown by top colour bar.

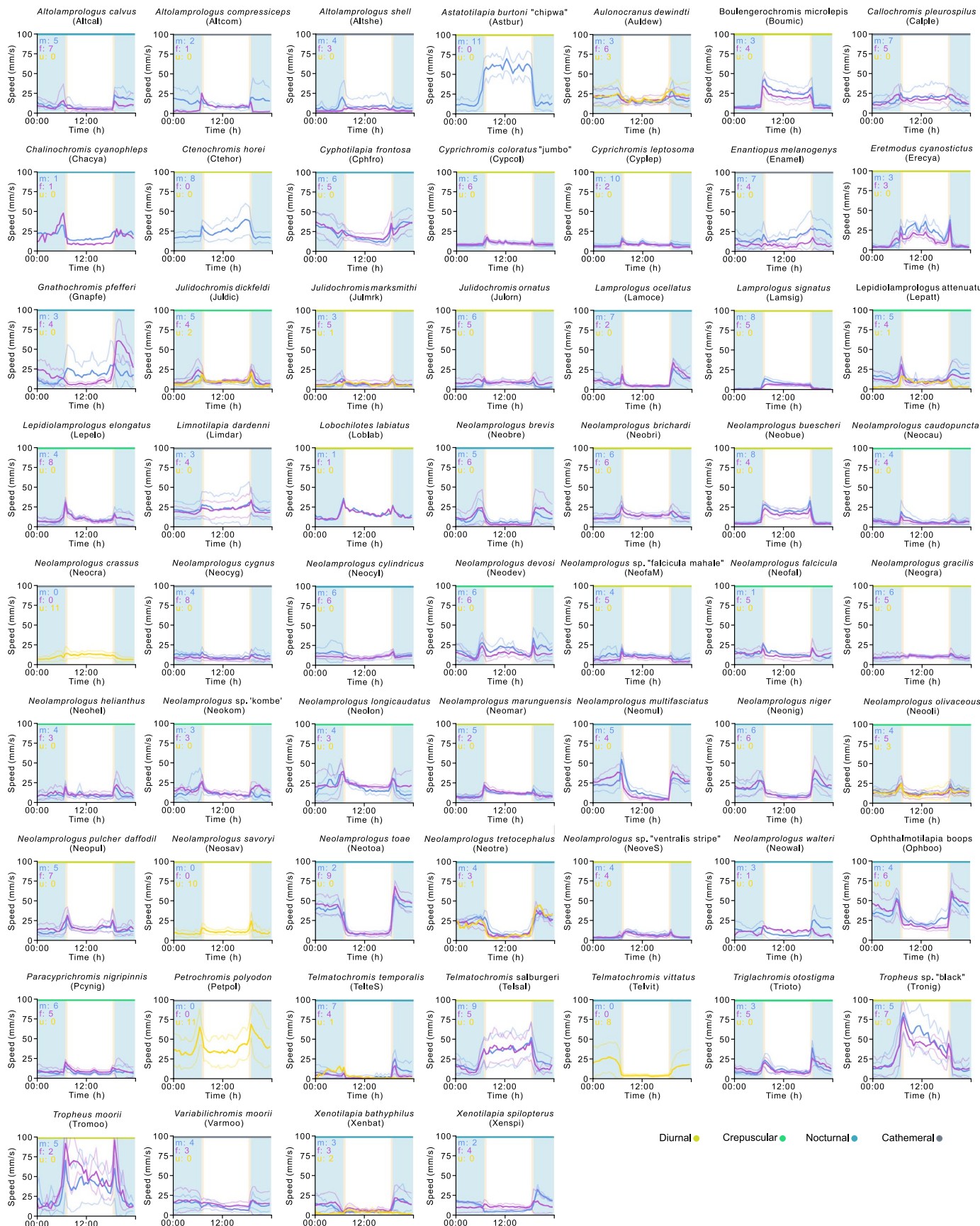

**Extended Data Fig. 2 | Daily speed for cichlid species in this study.** Daily speed (mean +/- SD) traces for 60 cichlid species separated by sex. Timeline and light cycle (12:12 h light:dark) is the same as in the schematic shown in Fig. 1b. m= male, f= female, u=undetermined (for cases where we could not accurately determine the sex). Number of animals assayed shown in the top left corner. Temporal guild shown by top colour bar.

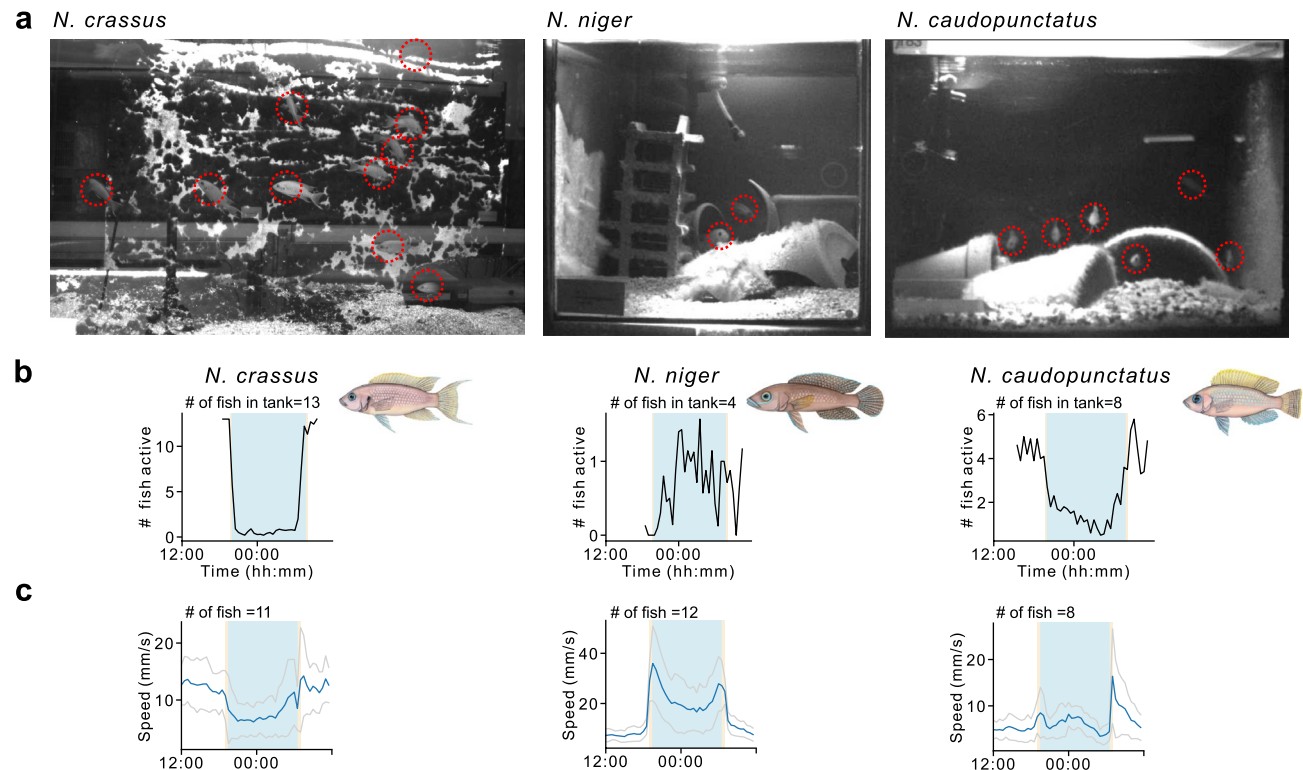

**Extended Data Fig. 3 | Home tank recordings. a**, Representative images of home tanks for each of the three species with observed fish highlighted by a red circle. **b**, number of active fish manually observed in home tanks for each species. **c**, daily speed (mean +/- SD) traces for three cichlid species *Neolamprologus crassus* (cathermal but with diurnality activity) *Neolamprologus niger* (nocturnal) *Neolamprologus caudopunctatus* (crepuscular). Number of animals assayed shown.

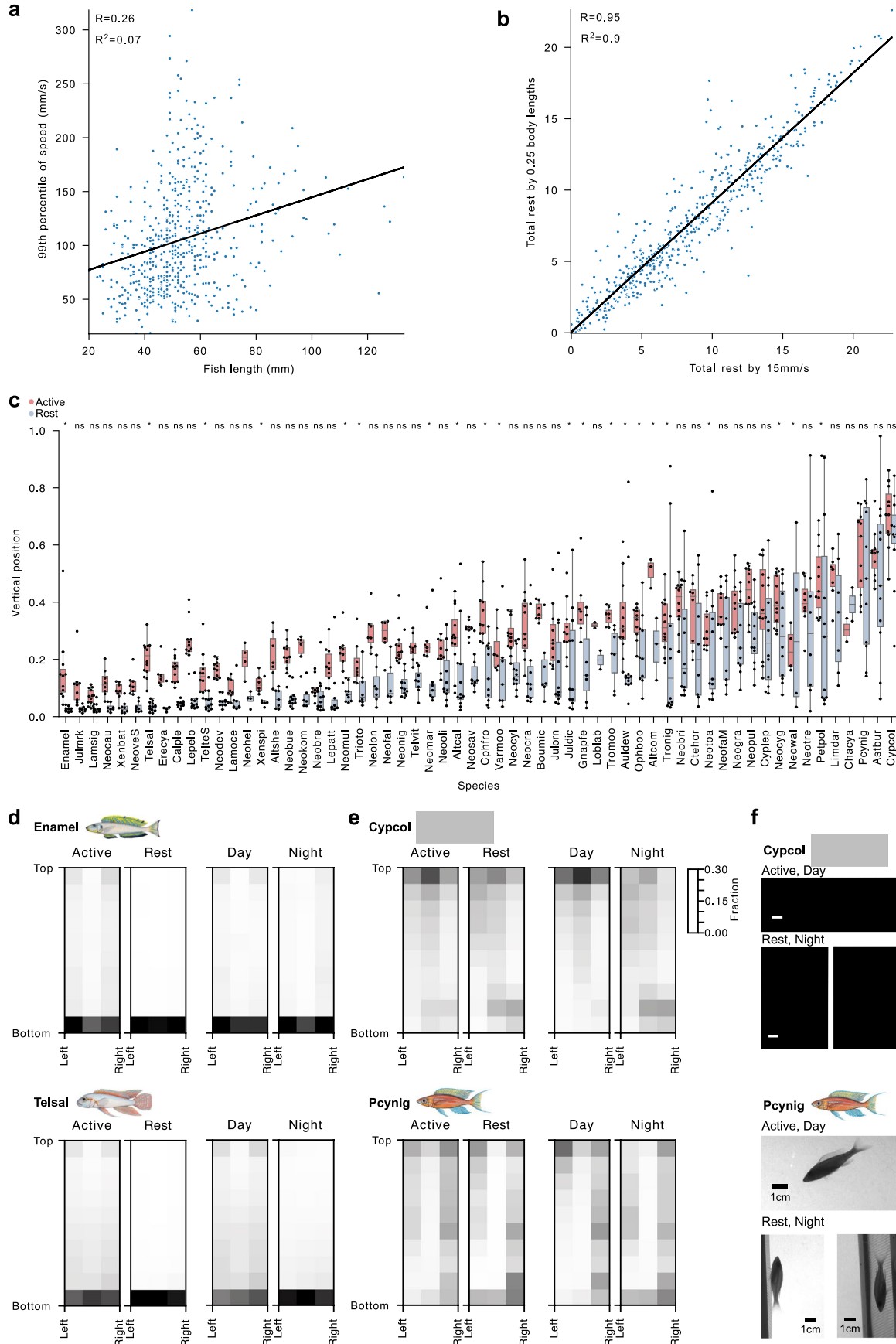

**Extended Data Fig. 4 | See next page for caption.**

**Extended Data Fig. 4 | Many species prefer to rest at the bottom of the tank.**
**a**, Fish length plotted against the 99th percentile of speed (mm/s) for each individual fish reveals little correlation between an individual fish's speed and body size. **b**, Rest defined by either an adaptive 0.25 body lengths per individual threshold correlates strongly with rest defined by an absolute 15 mm/s threshold does not drastically change the total rest. **c**, Average vertical position (scaled: 0: bottom, 1: top) during rest or active periods for each species, ordered by lowest vertical position during rest. For the majority of the cichlid species, rest bouts occurred predominantly when at the bottom of the tank, either just above, or in contact with the sandy substrate. Significance level of the Bonferroni corrected p-values for paired, two-sided t-tests of differences between mean vertical position of rest and active bouts are shown on top; * = p-value < 0.05. Each dot shows the average for one individual. The center line of each box represents the median, the box represents the 1st and 3rd quartiles (25th and 75th percentiles), and whiskers represent the full distribution excluding outliers. **d-e**, Density plots of positions for *E. melanogenys* (Enamel), *T. salzburgeri* (Telsal), *C. coloratus* (Cypcol) and *P. nigripinnis* (Pcynig) separated by Active and Rest or by Day and Night. **f**, Representative images of fish postures during Active phases of the Day, or Rest phases of the Night, for Cypcol and Pcynig. Species names are abbreviated using a six-letter code following Ronco et al. 2021 (Supplementary Data 1).

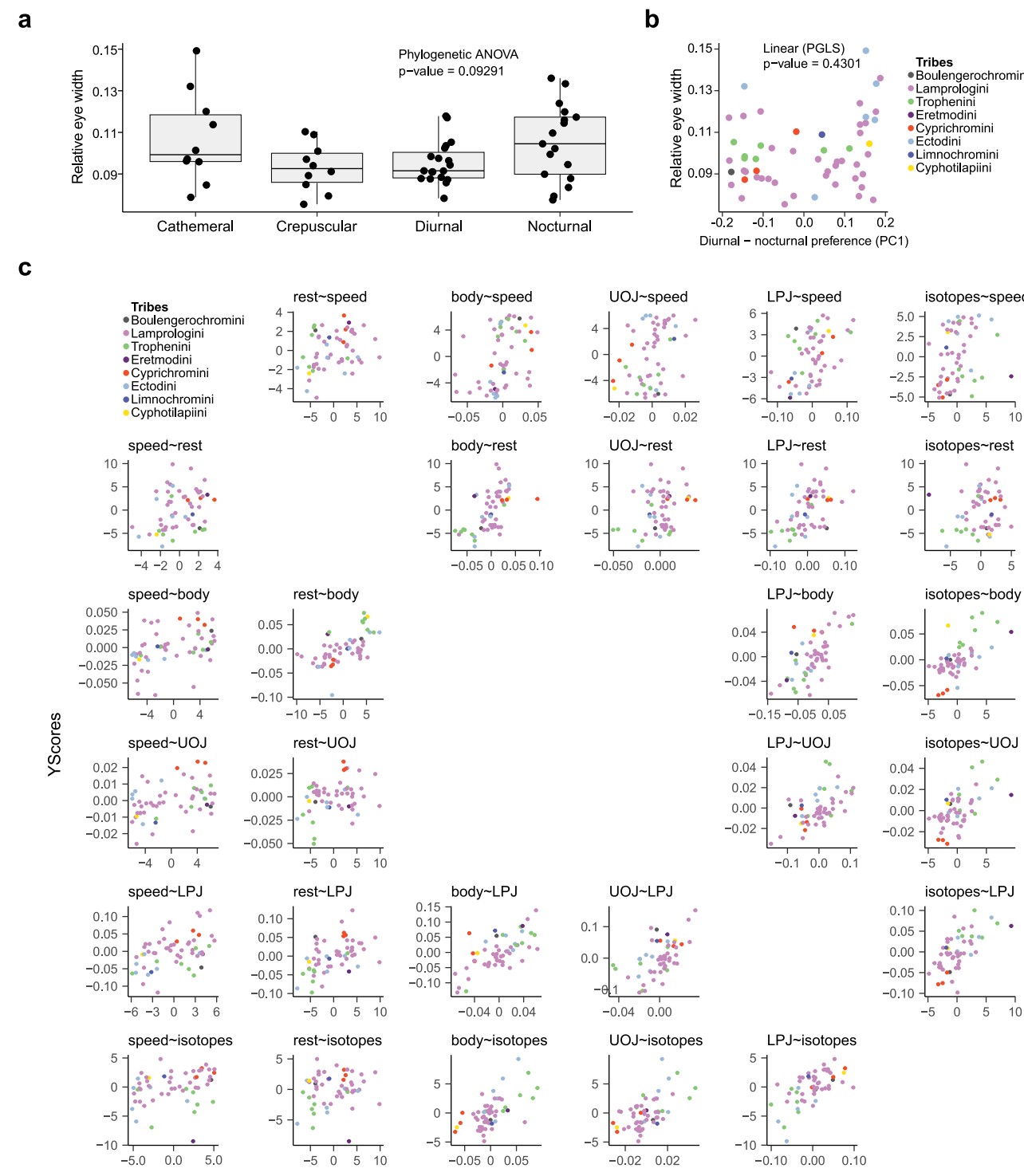

**Extended Data Fig. 5 | Relationships between temporal activity patterns and eco-morphological features of cichlids. a**, Boxplots of relative eye width (eye width divided by standard length) for cathemeral, crepuscular, diurnal, and nocturnal cichlid species. The center line of each box represents the median, the box represents the 1st and 3rd quartiles (25th and 75th percentiles), and whiskers represent maximal and minimal values. Inset displays statistics from a phylogenetically corrected ANOVA. **b**, Scatter plot of relative eye width plotted against PC1 loadings. Inset displays statistics from the fitting of a linear regression using phylogenetically corrected generalised least squares (pGLS). **c**, Pairwise relationships between PC1 loadings, PC2 loadings, total rest and published data for stable isotopes values and datasets of body and jaw morphology (UOJ: upper oral jaw; LPJ: lower pharyngeal jaw)[5]. Dots represent each species and are coloured by their tribe.

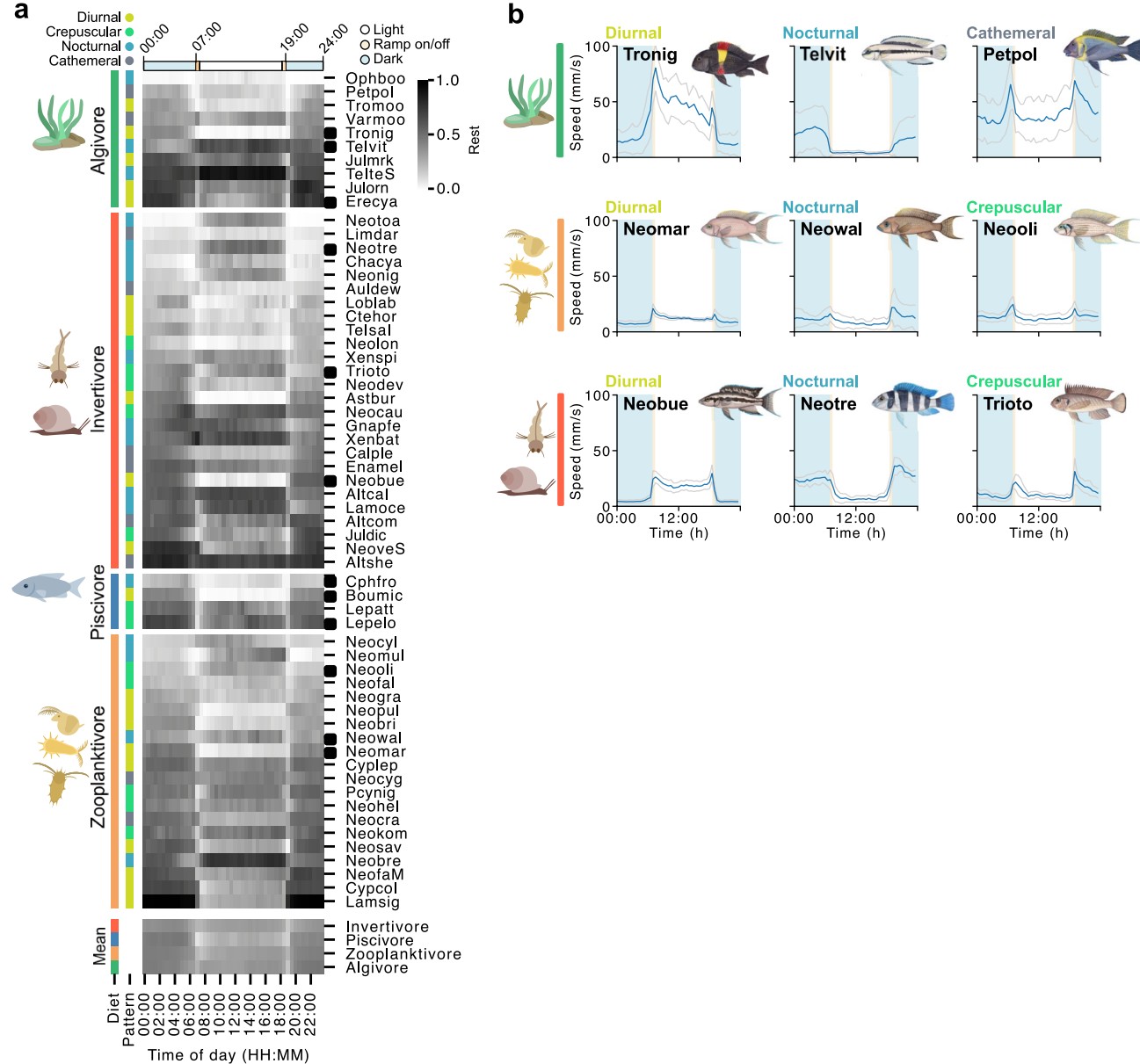

**Extended Data Fig. 6 | Ecological features and relationships to behavioural data. a**, top: heatmap of daily average of rest for each species grouped by diet guild. bottom: mean rest for each group. **b**, Examples of three species with diverse daily speed patterns per diet guild: Algivore, Invertivore, and Zooplanktivore, speed mean +/- SD. Species names are abbreviated using a six-letter code following Ronco et al. 2021 (Supplementary Data 1).

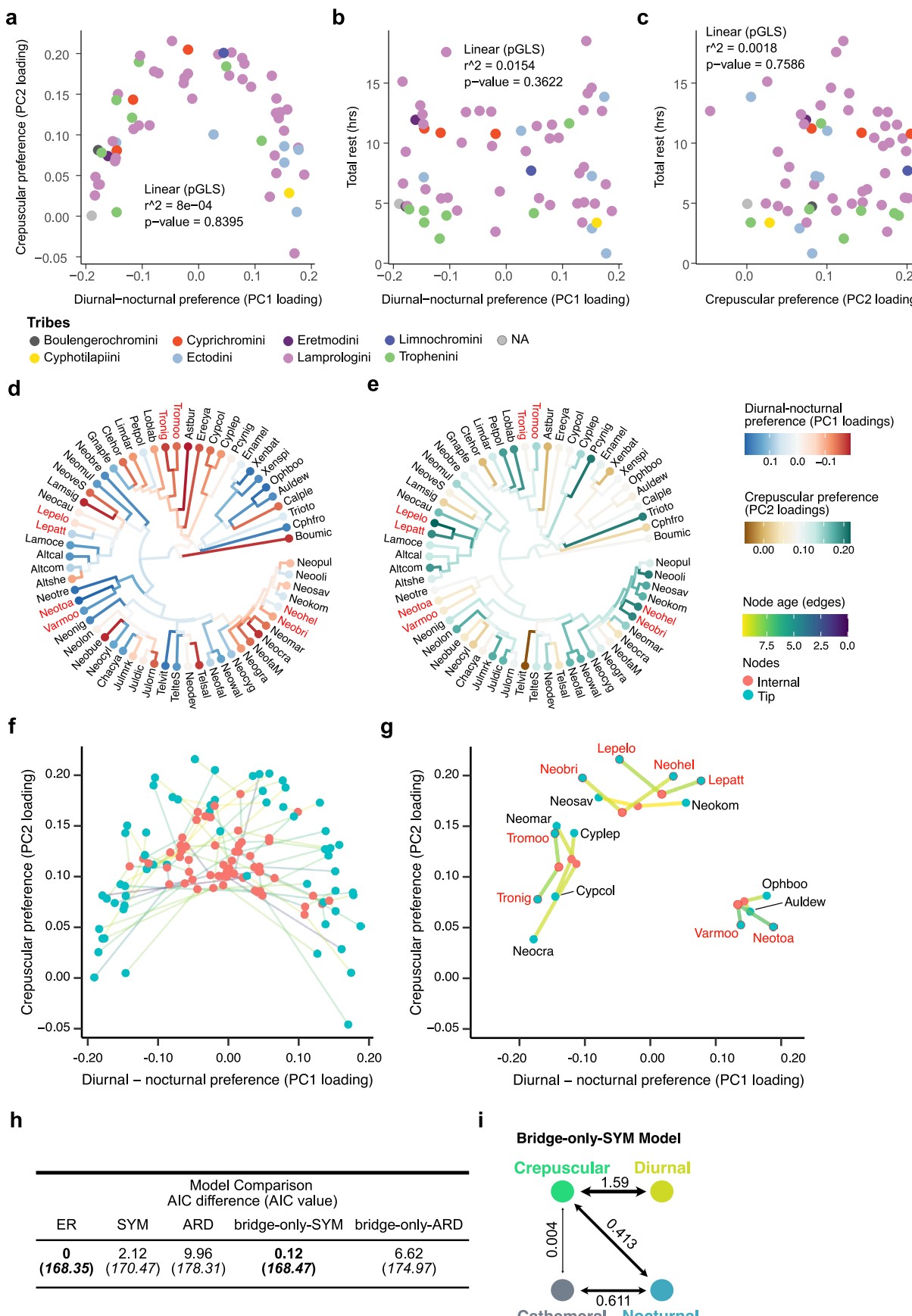

**Extended Data Fig. 7 | See next page for caption.**

**Extended Data Fig. 7 | Relationships between temporal activity patterns.**
**a-c**, The phylogenetically corrected pairwise relationships between PC1 loadings, PC2 loadings, and total rest, along with the statistics from the fitting of a linear regression using phylogenetically corrected generalised least squares (pGLS). Dots represent each species and are coloured by their tribe. **d-e**, Ancestral reconstructions of PC1 and PC2 loadings. Highlighted species codes correspond to highlights in panel g. **f**, Relationship between PC1 and PC2 loadings for both extant (blue dots) and ancestral nodes (red dots). Nodes are connected by their phylogenetic relationship, and edges are coloured by the age of the ancestral node. **g**, Subset of the data in panel f highlighting recent branching events with differing PC2 loading values reconstructed at their ancestral nodes. Nodes are connected by their phylogenetic relationship, and edges are coloured by the age of the ancestral node. **h**, Comparison of discrete model fits (AIC scores and delta AIC) for the evolution of temporal activity patterns. Both the ER and bridge-only-SYM models were significantly favoured over other models, but neither was favoured over the other. **i**, Transition rates between diurnal, nocturnal, crepuscular, and cathemeral states based on the best fitting bridge-only-SYM model (symmetric rates without direct transitions between diurnal and nocturnal states). The thickness of each line is representative, but not directly scaled to the rates between states.

**a**

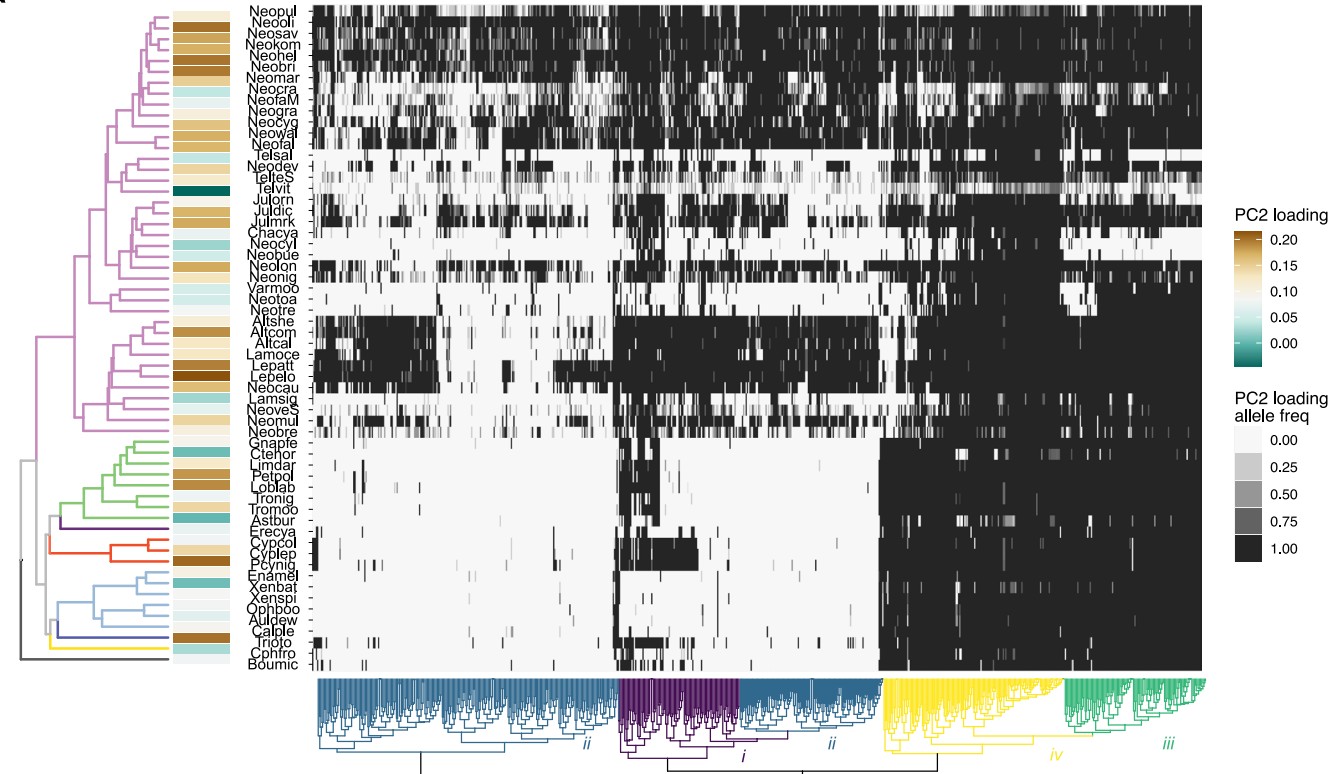

**b**

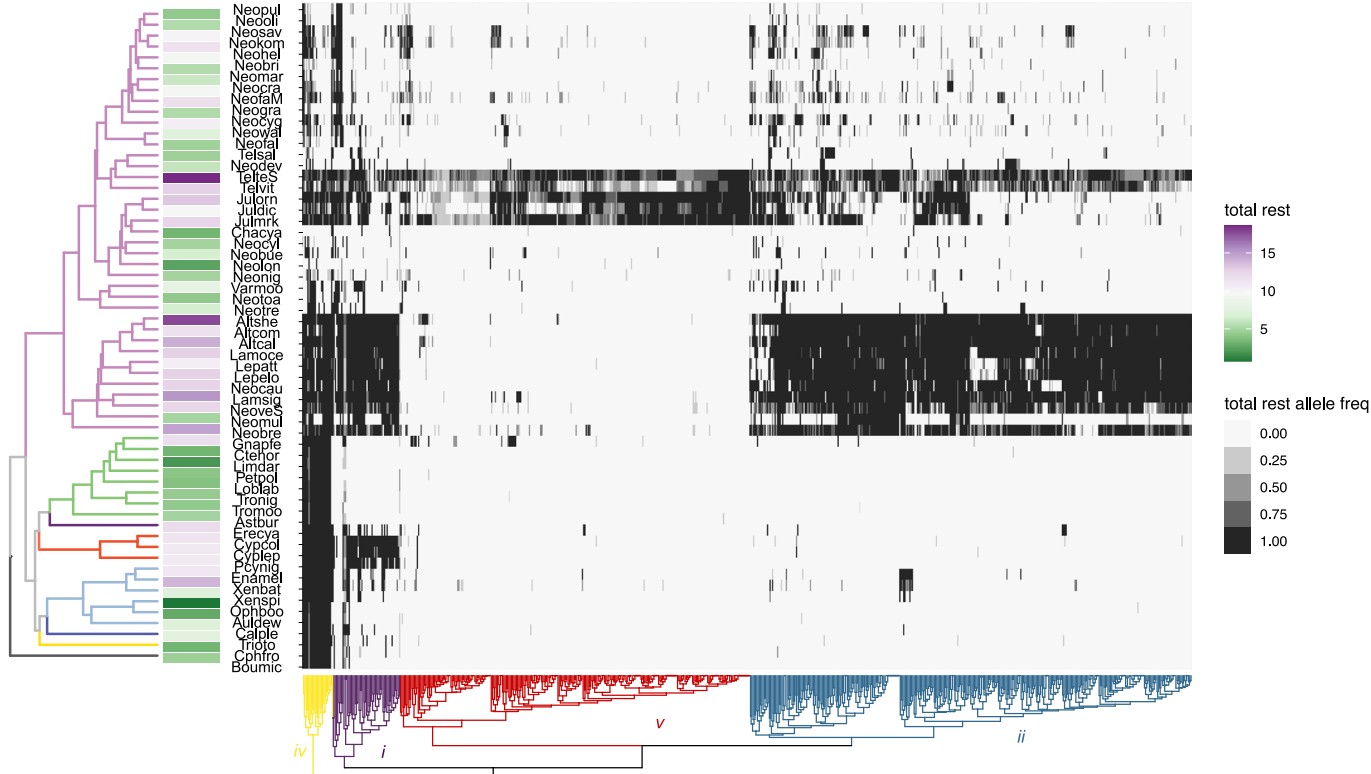

**Extended Data Fig. 8 | Clustering of highly associated variants for crepuscularity and total rest across cichlids. a,** Phylogenetic tree of the species in our dataset along with a heatmap of the allele frequencies of the high PC2 loading associated allele across all highly associated variants (HAVs) for PC2 loading (crepuscular preference). **b,** Phylogenetic tree of the species in our dataset along with a heatmap of the allele frequencies of the high total rest associated allele across all highly associated variants (HAVs) for total rest. Dendrograms represent relationships between HAVs. Clusters are coloured and annotated as in Fig. 3b.

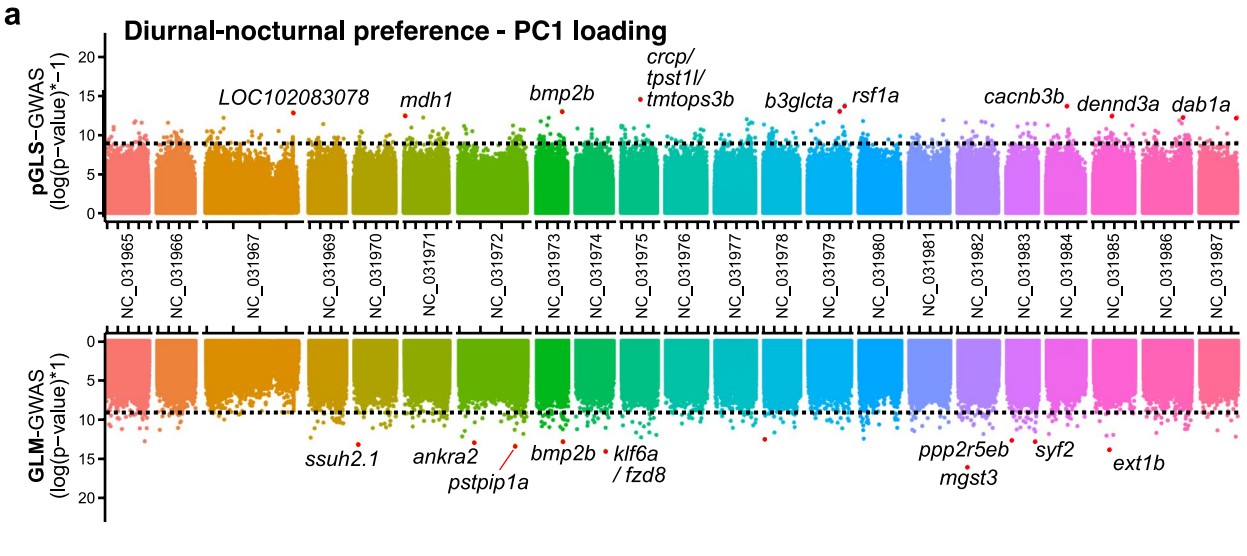

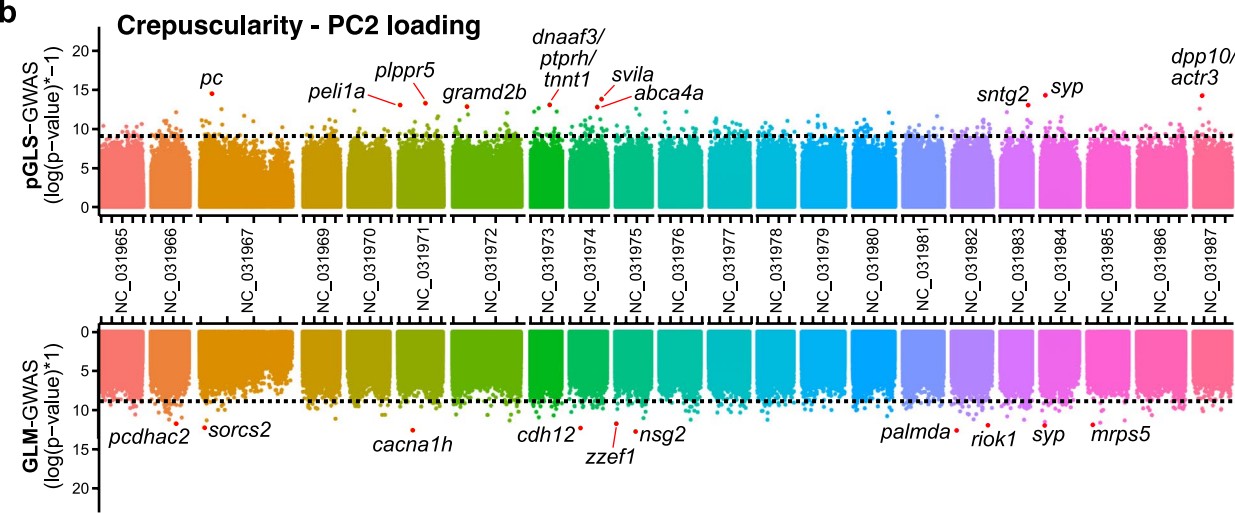

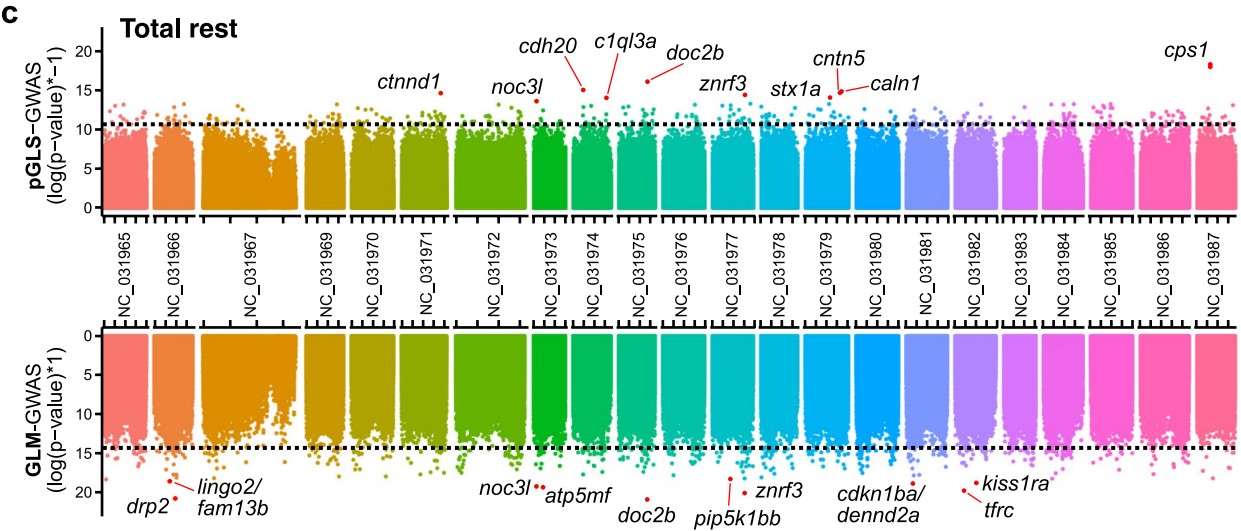

**Extended Data Fig. 9 | Variants associated with temporal activity patterns are distributed throughout the genome. a-c**, Manhattan plots showing the p-values for the association test for each SNP for both the pGLS and GLM tests for PC1 loadings (**a**), PC2 loadings (**b**), and total rest (**c**). Dotted lines indicated genome wide cutoffs for identification of HAVs. The top 10 HAV are annotated with the nearby gene(s). Statistics are derived from the fitting of a linear regression (GLM-GWAS), or linear regression using phylogenetically corrected generalised least squares (pGLS) (pGLS-GWAS).

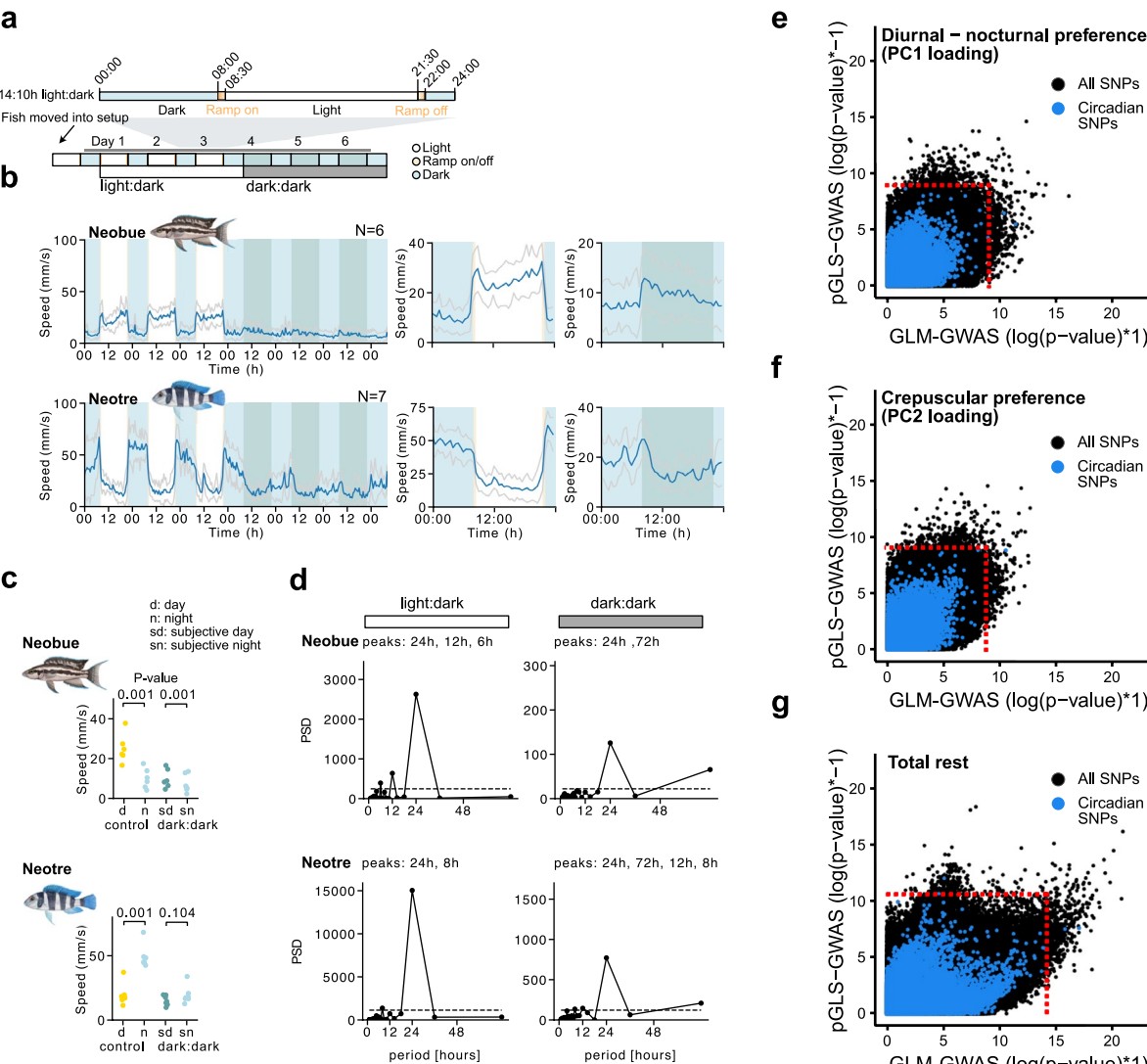

**Extended Data Fig. 10 | Daily activity patterns are regulated by internal circadian signalling and light. a**, Schematic of the timeline and light cycle for the behavioural assays of *N. buescheri* (Neobue) and *N. tretocephalus* (Neotre). Note that there is a 14:10 h light:dark cycle with 3 days of light:dark, and 3 days of dark:dark. **b**, left: weekly speed traces (mean +/- SD) of the full 6 days of assay, right: daily average traces (mean +/- SD) for the light:dark and dark:dark periods. **c**, quantification of speed during the day and night for light:dark, and subjective day and subjective night for the dark:dark period, P-values were calculated by a two-sided paired t-test. **d**, periodograms of the light:dark and dark:dark periods for both species. PSD = power spectral density. Significant peaks are listed on top of the plot for each condition. **e-g**, Scatter plots of the pGLS-GWAS p-value and GLM-GWAS p-values for all SNPs associated with PC1 loadings (e), PC2 loadings (f), and total rest (g). All SNPs associated with circadian genes are labelled. Dotted lines indicated genome wide cutoffs for identification of HAVs. Statistics are derived from the fitting of a linear regression (GLM-GWAS), or linear regression using phylogenetically corrected generalised least squares (pGLS) (pGLS-GWAS).

# Reporting Summary

## Statistics

For all statistical analyses, confirm that the following items are present in the figure legend, table legend, main text, or Methods section.

| n/a | Confirmed | |
|---|---|---|
| ☐ | ☒ | The exact sample size (*n*) for each experimental group/condition, given as a discrete number and unit of measurement |
| ☐ | ☒ | A statement on whether measurements were taken from distinct samples or whether the same sample was measured repeatedly |
| ☐ | ☒ | The statistical test(s) used AND whether they are one- or two-sided *Only common tests should be described solely by name; describe more complex techniques in the Methods section.* |
| ☐ | ☒ | A description of all covariates tested |
| ☐ | ☒ | A description of any assumptions or corrections, such as tests of normality and adjustment for multiple comparisons |
| ☐ | ☒ | A full description of the statistical parameters including central tendency (e.g. means) or other basic estimates (e.g. regression coefficient) AND variation (e.g. standard deviation) or associated estimates of uncertainty (e.g. confidence intervals) |
| ☐ | ☒ | For null hypothesis testing, the test statistic (e.g. *F*, *t*, *r*) with confidence intervals, effect sizes, degrees of freedom and *P* value noted *Give P values as exact values whenever suitable.* |
| ☒ | ☐ | For Bayesian analysis, information on the choice of priors and Markov chain Monte Carlo settings |
| ☒ | ☐ | For hierarchical and complex designs, identification of the appropriate level for tests and full reporting of outcomes |
| ☐ | ☒ | Estimates of effect sizes (e.g. Cohen's *d*, Pearson's *r*), indicating how they were calculated |

*Our web collection on statistics for biologists contains articles on many of the points above.*

## Software and code

Policy information about availability of computer code

| | |
|---|---|
| Data collection | Scripts for recording and tracking were written in python and are available online (https://github.com/annnic/cichlid-tracking). All scripts for performing quality control and running analysis of behavioural activity, including generation of plots of cichlid weekly and daily speeds were written in python and are available online (https://github.com/annnic/cichlid-analysis). |
| Data analysis | Scripts for analysis of eco-morphological data, construction of phylogenetic plots, highly associated variant analysis, and gene ontology analysis were written in R and available online (https://github.com/maxshafer/cichlid_sleep_gwas). Scripts for running genome wide association analysis, including the GATK python, generation of genome masks, and variant identification and filtering were written in bash and available online (https://github.com/maxshafer/cichlid_sleep_gwas). |

For manuscripts utilizing custom algorithms or software that are central to the research but not yet described in published literature, software must be made available to editors and reviewers. We strongly encourage code deposition in a community repository (e.g. GitHub). See the Nature Portfolio guidelines for submitting code & software for further information.

## Data

Policy information about availability of data

All manuscripts must include a data availability statement. This statement should provide the following information, where applicable:

- Accession codes, unique identifiers, or web links for publicly available datasets
- A description of any restrictions on data availability
- For clinical datasets or third party data, please ensure that the statement adheres to our policy

The time-calibrated species tree, morphology and stable carbon (C) and nitrogen and (N) isotope signatures were taken from Ronco et al. 20215 (data available on Dryad: https://datadryad.org/stash/dataset/doi:10.5061/dryad.9w0vt4bbf). Raw data from cichlid activity tracking is available as a supplemental file associated with this submission. The unbinned behavioural tracks are available on Dryad (https://datadryad.org/dataset/doi:10.5061/dryad.j0zpc86sv). Results from GWAS studies are available as supplemental file associated with this submission.

## Research involving human participants, their data, or biological material

Policy information about studies with human participants or human data. See also policy information about sex, gender (identity/presentation), and sexual orientation and race, ethnicity and racism.

| | |
|---|---|
| Reporting on sex and gender | *Use the terms sex (biological attribute) and gender (shaped by social and cultural circumstances) carefully in order to avoid confusing both terms. Indicate if findings apply to only one sex or gender; describe whether sex and gender were considered in study design; whether sex and/or gender was determined based on self-reporting or assigned and methods used.*<br>*Provide in the source data disaggregated sex and gender data, where this information has been collected, and if consent has been obtained for sharing of individual-level data; provide overall numbers in this Reporting Summary. Please state if this information has not been collected.*<br>*Report sex- and gender-based analyses where performed, justify reasons for lack of sex- and gender-based analysis.* |
| Reporting on race, ethnicity, or other socially relevant groupings | *Please specify the socially constructed or socially relevant categorization variable(s) used in your manuscript and explain why they were used. Please note that such variables should not be used as proxies for other socially constructed/relevant variables (for example, race or ethnicity should not be used as a proxy for socioeconomic status).*<br>*Provide clear definitions of the relevant terms used, how they were provided (by the participants/respondents, the researchers, or third parties), and the method(s) used to classify people into the different categories (e.g. self-report, census or administrative data, social media data, etc.)*<br>*Please provide details about how you controlled for confounding variables in your analyses.* |
| Population characteristics | *Describe the covariate-relevant population characteristics of the human research participants (e.g. age, genotypic information, past and current diagnosis and treatment categories). If you filled out the behavioural & social sciences study design questions and have nothing to add here, write "See above."* |
| Recruitment | *Describe how participants were recruited. Outline any potential self-selection bias or other biases that may be present and how these are likely to impact results.* |
| Ethics oversight | *Identify the organization(s) that approved the study protocol.* |

Note that full information on the approval of the study protocol must also be provided in the manuscript.

# Field-specific reporting

Please select the one below that is the best fit for your research. If you are not sure, read the appropriate sections before making your selection.

☒ Life sciences  ☐ Behavioural & social sciences  ☐ Ecological, evolutionary & environmental sciences

For a reference copy of the document with all sections, see nature.com/documents/nr-reporting-summary-flat.pdf

# Life sciences study design

All studies must disclose on these points even when the disclosure is negative.

| | |
|---|---|
| Sample size | Up to 14 adult individuals per species were tracked (average 9 individuals, with a range from 2-14 see Supplementary Data 1). A number of n = 12 is sufficient to correctly infer a 50% difference in the means of two groups with a relatively high standard deviation (1 SD = 43% of the mean) (80% power, 5% type I error). We expected that many of the parameters we measured through video tracking will require such power, as often behavioural phenotypes are quite variable between individuals, or will have small differences in mean measurements. Therefore, we chose to phenotype up to 14 individuals per species, or the maximum number of individuals available to us through breeding or purchase. |
| Data exclusions | Tracking for animals which became sick, died or escaped were excluded. |
| Replication | Given the large scale design and nature of our study, we did not incorporate mechanisms to measure the reproducibility of our findings. In some case, and for certain species, we were able to replicate their activity pattern, and in all of those cases our results were reproducible (for example, when one species was tested over two separate weeks, these results could be compared). |

| | |
|---|---|
| Randomization | Randomization was not relevant for our study design |
| Blinding | Experimenters were not blinded during data collection, however, most data analysis was done in an automated manner (online tracking of fish behaviour), and therefore blinding was not necessary. |

# Reporting for specific materials, systems and methods

We require information from authors about some types of materials, experimental systems and methods used in many studies. Here, indicate whether each material, system or method listed is relevant to your study. If you are not sure if a list item applies to your research, read the appropriate section before selecting a response.

## Materials & experimental systems

| n/a | Involved in the study |
|---|---|
| ☒ | ☐ Antibodies |
| ☒ | ☐ Eukaryotic cell lines |
| ☒ | ☐ Palaeontology and archaeology |
| ☐ | ☒ Animals and other organisms |
| ☒ | ☐ Clinical data |
| ☒ | ☐ Dual use research of concern |
| ☒ | ☐ Plants |

## Methods

| n/a | Involved in the study |
|---|---|
| ☒ | ☐ ChIP-seq |
| ☒ | ☐ Flow cytometry |
| ☒ | ☐ MRI-based neuroimaging |

## Animals and other research organisms

Policy information about studies involving animals; ARRIVE guidelines recommended for reporting animal research, and Sex and Gender in Research

| | |
|---|---|
| Laboratory animals | The full list of species used in our study is available in the Supplementary Information. Animals were all lab housed, and ultimately trace their origin back to wild caught in the Lake within a minimal number of generations. |
| Wild animals | N/A |
| Reporting on sex | Results are reported on the whole, and sex information is available in the supplementary data. |
| Field-collected samples | N/A |
| Ethics oversight | All experiments were performed under holding permit nrs. 1010H and 1035H, and experimental permit nrs. 2356 and 3102 issued by the cantonal veterinary office Basel. |

Note that full information on the approval of the study protocol must also be provided in the manuscript.

## Plants

| | |
|---|---|
| Seed stocks | N/A |
| Novel plant genotypes | N/A |
| Authentication | N/A |

