## [Peer Review File · Nature Ecology & Evolution]

Widespread temporal niche partitioning in an adaptive radiation of cichlid fishes

Corresponding Author: Professor Maxwell Shafer

Version 0:

Decision Letter:

21st August 2024

Dear Max,

Your Article, "Widespread temporal niche partitioning in an adaptive radiation of cichlid fishes" has now been seen by 3 reviewers. You will see from their comments copied below that while they find your work of considerable potential interest, they have raised quite substantial concerns that must be addressed. In light of these comments, we cannot accept the manuscript for publication, but would be very interested in considering a revised version that addresses these serious concerns.

In particular, please address the common concerns raised by Reviewers 1 and 3 about the categorization of temporal niches and potential effects of behavioural differences between fish species, for which Reviewer 1 has suggested additional experiments and phylogenetic analyses. Please also clarify all methodological issues related to the processing of genomic data, as highlighted by Reviewer 2 (whose comments you will find in the attached document).

We hope you will find the reviewers' comments useful as you decide how to proceed. If you wish to submit a substantially revised manuscript, please bear in mind that we will be reluctant to approach the reviewers again in the absence of major revisions.

If you choose to revise your manuscript taking into account all reviewer and editor comments, please highlight all changes in the manuscript text file, and include page and line numbers for the convenience of reviewers.

* Include a "Response to reviewers" document detailing, point-by-point, how you addressed each referee comment. If no action was taken to address a point, you must provide a compelling argument. This response will be sent back to the referees along with the revised manuscript.

* If you have not done so already we suggest that you begin to revise your manuscript so that it conforms to our Article format instructions at <http://www.nature.com/natecolevol/info/final-submission>. Refer also to any guidelines provided in this letter.

Link Redacted

If you wish to submit a suitably revised manuscript we would hope to receive it within 6 months. If you cannot send it within this time, please let us know. We will be happy to consider your revision so long as nothing similar has been accepted for publication at Nature Ecology & Evolution or published elsewhere.

Nature Ecology & Evolution is committed to improving transparency in authorship. As part of our efforts in this direction, we are now requesting that all authors identified as 'corresponding author' on published papers create and link their Open Researcher and Contributor Identifier (ORCID) with their account on the Manuscript Tracking System (MTS), prior to acceptance. This applies to primary research papers only. ORCID helps the scientific community achieve unambiguous attribution of all scholarly contributions. You can create and link your ORCID from the home page of the MTS by clicking on 'Modify my Springer Nature account'. For more information please visit www.springernature.com/orcid.

Thank you for the opportunity to review your work.

[redacted]

Reviewer expertise:

Reviewer #1: adaptive evolution, phylogenetics

Reviewer #2: genomics of animal behaviour

Reviewer #3: behavioural genomics in fish

Reviewers' comments:

Reviewer #1 (Remarks to the Author):

Here the authors perform behavioural assays on 60 different cichlid taxa from Lake Tanganyika and then examine the genetic basis for these behaviours.

The authors show differences among closely related Lake Tanganyika cichlids in their activity patterns with some active during the day, some active at night, and some most active at dawn and/or dusk. Rift lake cichlids are known to partition their habitats by depth and this manuscript also provides strong evidence for temporal niche partitioning that may have permitted the rapid adaptive radiations observed in this clade of fishes.

Additionally, the authors perform a GWAS analysis linking the behavioural data to genomic information. The GWAS analysis reveals few genes linked to the circadian clock and instead finds genes broadly implicated in neuronal and synaptic functions.

The manuscript is well-written and places the findings in an ecological and evolutionary context that centres around adaptive radiation. I have a few concerns that I outline below that I hope the authors can address. I list some major issues first, then some minor issues, alongside some specific comments.

Major Issues

Cichlids are social fish. The behavioural assay is based on work performed in cavefish populations which are not very social fish. I wonder how the signal may change if there are multiple fish in a tank that can directly interact. I would like the authors to confirm the patterns they are seeing are not a product of behavioural isolation. The authors should select a single representative species for each of the diurnal, nocturnal, and crepuscular behaviours (i.e., pick three species, do not do all 60) and see if the activity pattern remains constant. My assumption is that activity levels will increase overall, but it is important to ascertain if the general patterns remain. I am not familiar with the tracking software so I hope the program can deal with multiple (2-3) individuals.

I can't understand how you ultimately define whether a taxon belongs to the diurnal, nocturnal, or crepuscular behaviour groupings. It seems to be partially defined by the PCA (is that accurate?) but describe high variation across taxa, especially with respect to the crepuscular grouping. Is it possible to group these behaviours based on the PCA in an objective way?

Minor Comments

The authors propose how transitions among the different behaviours could occur, with crepuscular acting as a bridge between the diurnal and nocturnal states. Could the authors test these transition rates in a phylogenetic context (e.g., Price et al. 2014)?

Additionally, there are methods to compute phylogenetic signal for multivariate data – see Adams 2014 and Mitteroecker et al. 2024. The authors should consider these analyses as well.

Specific comments

Comparing behaviour to ecological features section – Unlike other sections, you do not mention the specific function for the PLS. I'm guessing two.b.pls and/or phylo.integration.

Figure 1g – Scale the branch lengths of your cluster analysis. The branches are really small, and I can't make out the groupings.

Figure 2a – painting the discrete behaviours here (as you did for Figure 1g) will help a reader visualize the distribution of the

behaviours across the tree.

Figure 2g – I do not believe there is a good reason to include both the corrected and non-corrected data here given the phylogenetic non-independence of your traits. Just include the corrected.

References cited in the review.

-Price S. A., Schmitz L., Oufiero C. E., Eytan R. I., Dornburg A., Smith W. L., Friedman M., Near T. J. and Wainwright P. C. 2014 Two waves of colonization straddling the K–Pg boundary formed the modern reef fish fauna. *Proc. R. Soc. B.* 281:2014032

-Adams, D. C., 2014, A Generalized K Statistic for Estimating Phylogenetic Signal from Shape and Other High-Dimensional Multivariate Data, *Systematic Biology*, 63, 5, Pages 685–697

-Mitteroecker, P., Collyer, M. L., Adams, D. C., 2024, Exploring Phylogenetic Signal in Multivariate Phenotypes by Maximizing Blomberg's K, *Systematic Biology*, syae035

Reviewer #3 (Remarks to the Author):

Thank you for the opportunity to review this manuscript. In it the authors present a comprehensive study on the understudied role of temporal niche partitioning in the widely investigated adaptive radiation of Lake Tanganyika cichlids. The authors obtained a comprehensive behavioral tracking dataset to monitor activity patterns of individuals from a large number of species over several days. Their data clearly points toward crucial differences across species in activity between light and darkness periods of study, as well as during periods with gradual changes in the light environment. They further integrate their data with readily available ecological and morphological data for these species, and conducted well executed genome-wide association studies that contribute with initial insights on molecular mechanisms underlying changes in daily activity patterns across species. Undeniably, this study provides an important contribution to further our understanding on the role of temporal niche diversification in this important system. I also believe their findings will be influential for future studies evaluating drivers of species diversification in this and other adaptive radiations, as well as across other systems.

I have however more reservations than those shown by the authors that their data provides direct evidence that temporal niche partitioning was crucial in the adaptive radiation of these fishes. The experimental design of the behavioral tracking dataset, while impressive, might unfortunately not be enough to capture ecologically-relevant changes in temporal activity across different light environments. I completely understand the practical decision to study activity of individuals while virtually isolated from social activity (with only access to visual and smell cues of very few conspecifics). Yet, the activity patterns observed from different species might be influenced by behavioral differences across these species in how they respond to this asocial situation. I would imagine that the fact that activity patterns across light environments were not correlated with any specific known morphological feature are quite unexpected, and might be indicative that the obtained behavioral data is capturing certain noise around values indicative of only circadian rhythms of these species. In this line, their results from genetic analyses found mostly variants previously associated with regulation of neural function, social behaviours and human disorders clearly associated with responses to social situations. While the authors provided a few examples from species in which observed temporal activity patterns matched expected physiological or ecological adaptations, I think a formal systematic comparison of their categorization with a trait known to vary strongly across diurnal-nocturnal species would strengthen the idea that they are capturing natural temporal activity patterns in their setup. I personally think those related to sensory systems should be ideal for this, as described in some of their examples.

Regarding the manuscript's presentation, I found it to be very well-written and easy to comprehend, with visually appealing figures.. As such, I only have a few additional comments regarding methods and potential additions to the manuscript:

- I think it would be beneficial to provide additional info on how fish were exactly fed during behavioural assays. I am surprised that the analyses did not detect a signal indicating peaks in activity associated with feeding times, as activity patterns typically increase during such periods.

- I suggest that the authors extend their discussion on correlations found/not found between behavioral data obtained and physiological measurements. How does their data align with the existing literature regarding the observed correlation between body size and total rest time?. Also, the lack of correlation between activity and body size seems contractor with a large body of literature in this matter.

- Methods:

o Regarding the methods to capture total rest in tracking data, should the movement threshold not be adjusted based on the size of each individual being tracked?

o As mentioned before, speed and body size seem to be strongly correlated across fishes. Therefore, I find it somewhat unusual that correlations with body size were calculated using previous data for each species without considering the body size of the individual participants in these experiments within their models.

Version 1:

Decision Letter:

10th April 2025

Dear Max,

Your manuscript entitled "Widespread temporal niche partitioning in an adaptive radiation of cichlid fishes" has now been seen by 2 reviewers, whose comments are attached. The reviewers have raised some remaining concerns which will need to be addressed before we can offer publication in Nature Ecology & Evolution. We will therefore need to see your responses to the criticisms raised and to some editorial concerns, along with a revised manuscript, before we can reach a final decision regarding publication.

We therefore invite you to revise your manuscript taking into account all reviewer and editor comments. When preparing the revision, please include analyses testing the fit of discrete character models as suggested by Reviewer #1, and add methodological and statistical details to the main manuscript as suggested Reviewer #3. Please highlight all changes in the revised manuscript file.

* If you have not done so already please begin to revise your manuscript so that it conforms to our Article format instructions at <http://www.nature.com/natecolevol/info/final-submission>. Refer also to any guidelines provided in this letter.

* Extended Data Figures - please ensure that any supplementary figures and tables that are crucial to the manuscript's conclusions are converted into Extended Data figures and tables to increase visibility of these data. Extended Data figures and tables are online-only (present in the online PDF and full-text HTML versions of the paper), peer-reviewed display items that provide essential background to the article but are not included in the main article due to space constraints. A maximum of ten Extended Data display items (figures and tables) is permitted.

Link Redacted

Nature Ecology & Evolution is committed to improving transparency in authorship. As part of our efforts in this direction, we are now requesting that all authors identified as 'corresponding author' on published papers create and link their Open Researcher and Contributor Identifier (ORCID) with their account on the Manuscript Tracking System (MTS), prior to acceptance. ORCID helps the scientific community achieve unambiguous attribution of all scholarly contributions. You can create and link your ORCID from the home page of the MTS by clicking on 'Modify my Springer Nature account'. For more information please visit www.springernature.com/orcid.

[redacted]

Reviewers' comments:

Reviewer #1 (Remarks to the Author):

Here the authors perform behavioural assays on 60 different cichlid taxa from Lake Tanganyika and then examine the genetic basis for these behaviours.

This is the second time I have seen this manuscript, and it is exciting to see how the work has progressed. While almost all my comments were addressed, I would still like to see some empirical data on the 'crepuscular bridge' hypothesis. The authors added an ancestral state reconstruction and tested various evolutionary models against their activity data but I have some minor concerns around how these were conducted.

I'm happy to see the authors discuss in their rebuttal the pilot study on the group assessments. I agree that maintaining the health of our study organisms should always be paramount.

The ancestral state reconstruction gets to some of the questions around activity transitions, although I will caution about over-interpreting results from these analyses. See Holland et al. 2020 for details on this. The larger issue was the evolutionary model analysis – I don't understand how these data can lead to evidence for or against the 'crepuscular bridge' hypothesis. You also don't assess the fit of your PC activity data to a multi-peak model, whereby peaks correspond to diurnal, crepuscular, nocturnal, cathemeral patterns - this would probably give you the best sense for transition rates. I don't believe this continuous model fitting analysis adds much to the story and could be removed.

I would however recommend using the routines in Liam Revell's phytools package – specifically the fitMK family (the help file and included examples are exceptional) – where you can compare the fit of discrete character models. This should give you a better sense for the transition rates among temporal activity patterns. You can then test whether your data fit a model of equal vs. different transition rates. This should allow you to more directly compare rates and make inferences about the 'crepuscular bridge' hypothesis.

Line by line comments

408-412: Can you provide more clarity here? I became lost around the "tissue-specific clocks" line. Is this saying that different tissues and cells can exhibit different rhythms and respond differently at different times of the day?

495: With the removal of this line, I believe there is no mention of how sex is treated in the manuscript. There should be a note about whether sex was considered to impact activity patterns.

590-594: What would constitute a superior model? Typically set to be around 2 AICc units away from the next best-competing model (Burnham and Anderson 2002). Could present the delta AICc values to make this clear. The fitMK analysis should also include the delta AICc threshold, i.e., that 2-4 units from next competing model is typically considered to demonstrate superior model fit.

References mentioned in the review.

Burnham KP, Anderson DR (2002) Model selection and multimodel inference: a practical information-theoretic approach, 2nd edn. Springer, New York

Holland, B.R., Ketelaar-Jones, S., O'Mara, A.R. et al. Accuracy of ancestral state reconstruction for non-neutral traits. *Sci Rep* 10, 7644 (2020). <https://doi.org/10.1038/s41598-020-64647-4>

Revell L. J. (2024). phytools 2.0: an updated R ecosystem for phylogenetic comparative methods (and other things). *PeerJ*, 12, e16505. <https://doi.org/10.7717/peerj.16505>

Reviewer #1 (Remarks on code availability):

GWAS looks good - code separated into chunks to allow a user to follow each step - file names are informative. Comments present throughout.

ReadME is present - links to preprint.

I did not attempt to run the code.

Reviewer #3 (Remarks to the Author):

I commend the authors for their revision of the manuscript and congratulate them once again on an interesting study with a good scientific approach and excellent visualizations. This revised version elegantly refines the text without abrupt modifications or major additions, incorporating most of the reviewers' suggestions. In my view, these edits have strengthened the manuscript by enhancing clarity, appropriately tempering the direct conclusions drawn from the findings, and clearly distinguishing speculative aspects in the discussion.

I believe there are still areas where additional improvements could further refine the manuscript. Below, I outline my comments, focusing on the authors' responses to my previous feedback as well as revisions prompted by other reviewers' suggestions.

In addition to minor comments that I consider sufficiently addressed, I have grouped my previous suggestions for improvement into three main categories:

1. The relevance of testing species in isolation.

In response to Reviewer 1's suggestions, the authors present new data (included only in the response letter) on activity patterns in three species studied in a more ecologically relevant social context. Given that current editorial policies make response letters publicly available, I understand the authors' rationale for keeping this data separate. However, I respectfully disagree with their decision to exclude it from the manuscript.

This information is important for contextualizing the study's methods and understanding its limitations. As I understand it, the feasibility of conducting experiments in a fully naturalistic setup is constrained by logistical challenges and the difficulty of developing a unified approach for species with significant biological differences. The authors' previous data suggest that

individual activity patterns in isolation are representative of those in ecologically relevant conditions. However, the response letter lacks details on species selection criteria and key methodological aspects of data collection. If space allows, I strongly recommend incorporating this information into the main text or extended data

While I acknowledge the authors' reasoning, I remain unconvinced that testing fish in isolation does not introduce noise affecting observed activity patterns and the genetic variants associated with them. The addition of the term "social behavior" when discussing the genetic findings does not, in my view, sufficiently acknowledge this limitation.

2. Further comparisons of morphological traits between diurnal and nocturnal species.

The inclusion of correlations between morphological traits and temporal activity patterns is well executed and a valuable addition to the manuscript.

3. Considerations for tracking data validity in species with varying morphology and feeding behaviors.

I find this concern adequately addressed with the incorporation of an alternative approach to estimate body size and a clearer description of feeding patterns during testing. I find concerns additionally raised by reviewer 2 in this regard are also adequately addressed.

Additionally, I have some comments based on responses to other reviewers. I found that their comments aligned with mine, and I appreciate the authors' efforts to revise the manuscript accordingly.

All reviewers emphasized the need for a more nuanced presentation of results related to the timeline of temporal activity adaptations and transitions between behavioral types. While I am not an expert in the specific analytical methods suggested by the other reviewers, I believe the authors have addressed these concerns well by incorporating phylogenetic information and refining their interpretations throughout the manuscript.

Regarding the analyses linking genes and gene categories to temporal activity patterns, I did not raise concerns in my initial review, but I now provide additional comments in light of the responses to Reviewer 2 and my own experience with similar analyses:

- SNP-gene associations: The authors apply standard methods using default settings, but I agree that these should be more explicitly stated in the manuscript to ensure clarity for readers unfamiliar with snpEff or SNP2GO. Specifically, it should be clear how genes and GO terms are assigned to SNPs and when these assignments are used for downstream analyses. I was unaware that neuronal genes tend to have larger intergenic regions than the median, but I do not believe this introduces a bias in the authors' analyses, which focus on overall associations with gene types and functional categories. The revised manuscript appropriately clarifies that no causal links can be inferred from these findings.
- INDEL analyses: In response to Reviewer 2 and in the manuscript, the authors partially justify not including INDEL analyses by stating that such analyses are increasingly difficult. While I did not find the absence of INDEL analyses problematic given the study's broader focus, I note that recent methodological advances have made these analyses more accessible.
- Tests of allele stratification and behavioral phenotypes: There appears to be a misunderstanding regarding the statistical validation of allele distributions between clades. The authors claim significant differences in allele stratification (line 293), but as far as I can tell, these results are not based on formal statistical tests. If such tests have been performed, the methods and results should clarify this. If not, I recommend conducting the appropriate statistical tests.

Overall, I appreciate the authors' revisions and believe these final clarifications would further strengthen the manuscript.

*****END*****

Version 2:

Decision Letter:

3rd June 2025

Dear Max,

Thank you for submitting your revised manuscript "Widespread temporal niche partitioning in an adaptive radiation of cichlid fishes" (NATECOLEVOL-24061572B). It has now been seen again by one of the original reviewers and their comments are below. The reviewer finds that the paper has improved in revision, and therefore we'll be happy in principle to publish it in Nature Ecology & Evolution, pending minor revisions to satisfy the reviewer's final requests and to comply with our editorial and formatting guidelines.

If you have not done so already, please ensure that you also email us completed copies of the Reporting summary and Editorial policy checklists:

Reporting summary: https://www.nature.com/documents/nr-reporting-summary.pdf

Editorial policy checklist: https://www.nature.com/documents/nr-editorial-policy-checklist.pdf

Once we receive the file in an editable format, we will perform detailed checks on your paper and will send you a checklist listing our editorial and formatting requirements. Please do not upload the final materials or make any revisions until you receive this checklist from us. However you can start updating your code as requested by the referee.

[redacted]

Reviewer #1 (Remarks to the Author):

Here the authors perform behavioural assays on 60 different cichlid taxa from Lake Tanganyika and then examine the genetic basis for these behaviours.

This is the third time I have seen this manuscript, and the authors have made some excellent additions to their paper. I am happy to see an expansion of the model fitting analysis and the incorporation of the pilot study to the main text.

I have a few minor comments for the authors that I list below.

Line 253-266 and Extended data figure 7h. You cite an AIC score of 2 units to distinguish a model from other competing models – the ER model (AIC = 168.35) and Bridge SYM model (AIC = 168.47) exhibit similar support. You need to add a caveat in the main text that the ER model also garnered high support alongside the Bridge SYM model. Similarly, in panel h of Extended data figure 7 the delta AIC for the ER model should be 0 rather than -0.12 as this model exhibited the lowest AIC score, and you should subtract this AIC value from the other model AIC values to gain the delta AIC (i.e., Bridge SYM would be $dAIC = 0.12$ etc.).

Line 431. Add a definition for 'zeitgebers' given the broader readership of this journal.

Line 745. Please add new scripts generated as part of these reviews to GitHub to aid reproducibility.

<https://github.com/annnic/cichlid-analysis>

Reviewer #1 (Remarks on code availability):

As discussed in the Comments to Author section, additional analyses were added to the manuscript and should be included in the GitHub repository. In particular -> <https://github.com/annnic/cichlid-analysis>

Summary:

We thank the reviewers for taking the time to read our manuscript and provide useful feedback on our study. We appreciate the positive comments from all three reviewers on the quality of our work, the impact of our study on our understanding of adaptive radiations, and the constructive feedback and suggestions they have provided. In the revised version, we have fully addressed all of their comments. In particular we have now provided:

- 1) An analysis and ancestral reconstruction of temporal activity patterns that support our hypothesis of a crepuscular bridge between nocturnal and diurnal activity patterns.
- 2) An analysis of correlations between our metrics (total rest, diurnal/nocturnal preference, crepuscularity) and both body size and eye size across species.
- 3) A clarification of our methodology, particularly as it pertains to the genomic studies, our classification of temporal activity niches, and analysis of the influence of body size on our measurements.
- 4) An expanded discussion of the limitations and interpretations of our study, including alternative hypotheses that could explain our results regarding transitions between temporal activity niches, and genomic associations
- 5) Inclusion of analysis in the response to reviewers of the activity patterns of several representative species in more social contexts, which match the activity patterns observed in our reductionist setup.

Please see below for a detailed point-by-point response to each reviewers' comment. We look forward to your further feedback.

Reviewer #1 (Remarks to the Author):

Here the authors perform behavioural assays on 60 different cichlid taxa from Lake Tanganyika and then examine the genetic basis for these behaviours.

The authors show differences among closely related Lake Tanganyika cichlids in their activity patterns with some active during the day, some active at night, and some most active at dawn and/or dusk. Rift lake cichlids are known to partition their habitats by depth and this manuscript also provides strong evidence for temporal niche partitioning that may have permitted the rapid adaptive radiations observed in this clade of fishes.

Additionally, the authors perform a GWAS analysis linking the behavioural data to genomic information. The GWAS analysis reveals few genes linked to the circadian clock and instead finds genes broadly implicated in neuronal and synaptic functions.

The manuscript is well-written and places the findings in an ecological and evolutionary context that centres around adaptive radiation. I have a few concerns that I outline below that I hope the authors can address. I list some major issues first, then some minor issues, alongside some specific comments.

We thank the reviewer for their positive comments on the manuscript and our findings, and their constructive criticisms. We believe we have fully addressed their comments, with specific details highlighted in the sections below. In particular, we discuss a pilot study that used video tracking to analyze activity profiles of several cichlid species in social contexts. These results address whether sociality influences activity patterns and includes example diurnal, nocturnal, and crepuscular species. As suggested by the reviewer, we now also include multivariate methods and ancestral reconstructions, both of which support our previous conclusions.

Major Issues

R1.1 Cichlids are social fish. The behavioural assay is based on work performed in cavefish populations which are not very social fish. I wonder how the signal may change if there are multiple fish in a tank that can directly interact. I would like the authors to confirm the patterns they are seeing are not a product of behavioural isolation. The authors should select a single representative species for each of the diurnal, nocturnal, and crepuscular behaviours (i.e., pick three species, do not do all 60) and see if the activity pattern remains constant. My assumption is that activity levels will increase overall, but it is important to ascertain if the general patterns remain. I am not familiar with the tracking software so I hope the program can deal with multiple (2-3) individuals.

We thank the reviewer for this comment. We agree that many cichlid species from Lake Tanganyika live in social groups of varying stability and size, and that these interactions could have an effect on their activity patterns. However, this is not the case for all species. Many of the cichlid species we tested are aggressive towards conspecifics and compete for shelters and breeding territories when in a novel environment, and many species live solitary lives, or live in harems or other sex-biased groupings.

Before we began this project, we performed pilot studies, where we filmed several species of cichlids overnight in their home tanks. This included species we later confirmed to be diurnal (*Neolamprologus crassus*), nocturnal (*N. niger*), and crepuscular (*N. caudopunctatus*). These species are group breeders, and live in either multi-generational groups, or as harems

(1 male, multiple females). These experiments revealed significant differences in activity patterns, and we have subsequently quantified these videos (**Response to Reviewers Figure 1**). These species were recorded in the presence of additional environmental enrichment, as well as multiple conspecifics that had already formed stable social groups. Because of the nature of home tanks we were unable to record for more than 16 hours or use our tracking program. Instead we opted to manually track activity and recorded for 20 frames/hour whether or not visible fish were moving. Non-visible fish were assumed to be motionless in their hiding places. In all 3 cases, these species displayed similar preferences for activity timing (diurnal, nocturnal, crepuscular) as that observed in our single-fish setup.

In the experiments that we present in this manuscript, we opted for the current ‘reductionist’ experimental design due to both logistical reasons (related to the difficulty of tracking multiple individuals, or using different experimental setups per species), but also for biological and experimental reasons. For example, there are ethical concerns over housing fish in the same tank that could fight and kill each other for territories. Our current setup allowed all species to be tested in the same reductionist paradigm, avoiding complications due to differing sociality. Importantly, our setup still allowed some social interactions if the fish chose to do so, including visual, auditory, and olfactory signalling (through the mesh dividers), but in a much higher throughput and more quantitative way than our pilot experiments. Many other biotic and abiotic factors were purposefully omitted to allow for this reductionist approach. These include interspecies interactions (predator or prey species), and many abiotic factors present in the lake, but not in aquaria (rocks, plants, weather patterns, moon cycles). The effects of these factors on the circadian activity and inactivity patterns of cichlids could be addressed in follow up studies.

Response to Reviewers Figure 1. Top row: midnight centred daily activity plots of three species (mean +/- standard deviation). Bottom row: observations of number of active fish in home tank settings.

R1.2 Q: I can't understand how you ultimately define whether a taxon belongs to the diurnal, nocturnal, or crepuscular behaviour groupings. It seems to be partially defined by the PCA

(is that accurate?) but describe high variation across taxa, especially with respect to the crepuscular grouping. Is it possible to group these behaviours based on the PCA in an objective way?

We apologize for any confusion on the methods we used to categorize species into temporal activity patterns. We have changed **Figure 1** to allow readers to better understand the groupings, and added clarification to the text (**line 538**). As we mentioned in the original version of the manuscript, the principle components analysis (PCA) is only performed on the daily averages for each species, and does not include information on within-species variability (differences between individual fish within a species). We observed that some species in our dataset had large intraspecific variability (see **Extended Data Fig. 1**), and we felt the species activity types were misrepresented by only using the PCA measures. Therefore, we added a second layer to our classification to identify species with high variability as potentially arrhythmic or cathemeral. In this classification, species where the min to max daily average 30 minute binned speed difference was smaller than two times the average mean standard deviations of those, were classified as cathemeral (see **Fig. 1g**). For example, *Callochromis pleurospilus* in **Fig. 1H**, and **Extended Data Fig. 1**, was categorized as diurnal by PC analysis, but because it had such high variation between individuals, during the second level classification, it was categorized as cathemeral. The PCA clustering to define diurnal/nocturnal/crepuscular groupings and the cutoff for cathemerality are both objective measures. In addition, we have extended the branch lengths for **Figure 1g** and added more detail to the text to make this clearer to our readers (**line 541**).

Minor Comments

R1.3 The authors propose how transitions among the different behaviours could occur, with crepuscular acting as a bridge between the diurnal and nocturnal states. Could the authors test these transition rates in a phylogenetic context (e.g., Price et al. 2014)?

Additionally, there are methods to compute phylogenetic signal for multivariate data – see Adams 2014 and Mitteroecker et al. 2024. The authors should consider these analyses as well.

We thank the reviewer for this comment, as well as the similar comment raised by reviewer #2- see comments **R2.1** and **R2.4**. Ideally we could test these hypotheses in a phylogenetic context, to determine the tempo in the evolution of crepuscularity and diurnal/nocturnal preference, the relative timing in the evolution of both, and examine the reconstructed states at internal nodes representing common ancestors for species with opposite diurnal and nocturnal activity preferences. For example, in a recent paper by some of us (Ronco et al., 2021, Nature), ancestral reconstructions were performed for several morphological traits using a dataset containing information for virtually all of approximately 240 cichlid species in Lake Tanganyika, and this provided evidence for staggered bursts in the evolution of each trait.

In response to this point and a similar point raised by Reviewer #2 (**R2.4**), we now provide analyses and ancestral reconstructions for diurnal/nocturnal and crepuscular preferences. We tested multiple evolutionary models for both diurnal/nocturnal preference (PC1 loadings) and crepuscular preference (PC2 loadings), including brownian motion (BM), Ornstein–Uhlenbeck (OU), and early-burst (EB) models. AICc scores did not differ substantially between models (difference was < 2), suggesting that no model was favoured over the simpler

BM model. Therefore, we used separate BM models to reconstruct ancestral states for PC1 and PC2 (**Extended Data Figure 5d-e**). As before, we observed a parabolic relationship between PC1 and PC2 loadings when considering both extant and ancestral nodes, and that species only occupy trait space along a bridge-like continuum (**Extended Data Figure 5f**).

Overlaying the phylogenetic tree structure by connecting parental nodes to daughter nodes with edges, we observed that the majority of connections between nodes mirrored the bridge-like continuum. Furthermore, when we examined the most recent branching events, we observed that ancestral nodes with higher PC2 values which branch into extant species with divergent PC1 values (e.g. *N. brichardi* vs *N. helianthus*, or *Lepidilamprologus elongatus* vs *L. attenuatus*). This is in contrast to ancestral nodes with lower PC2 values, which branch into extant species with similar PC1 values (e.g. diurnal *Tropheops moori* vs *T. niger*, or nocturnal *N. toae* vs *Variabilichromis moorii*) (**Extended Data Figure 5g**). However, there are limitations to this analysis, and we provide some caveats in the text. For example, the taxonomic coverage of our dataset is more limited than in Ronco et. al., 2021, and our dataset lacks monophyletic sister species with opposite diurnal/nocturnal preferences, and is missing representation from some clades that make interpretation of early branching events difficult.

In addition, we have added discussion of the limitations of our interpretations of the crepuscular bridge model, and highlight alternative hypotheses on evolutionary transitions between activity patterns in the discussion (**lines 379-395 & 396-419**).

As suggested, we have also included a calculation of a multivariate K metric for the raw speed data using the function `physignal` from the R package `geomorph`¹. The result of this ($K_{\text{multi}} = 0.411$) is consistent with our previous analyses of Bloomberg's K for PC1, PC2, or total rest (**line 162**).

Specific comments

R1.4 Comparing behaviour to ecological features section – Unlike other sections, you do not mention the specific function for the PLS. I'm guessing `two.b.pls` and/or `phylo.integration`.

This is correct, we used the `two.b.pls` function. We have added this to the text (**line 579**).

R1.5 Figure 1g – Scale the branch lengths of your cluster analysis. The branches are really small, and I can't make out the groupings.

We have made the branch lengths longer in **Figure 1g**, and this helps the reader to better understand the temporal pattern groupings.

R1.6 Figure 2a – painting the discrete behaviours here (as you did for Figure 1g) will help a reader visualize the distribution of the behaviours across the tree.

We thank the reviewer for this useful comment and have added the discrete behaviours onto **Figure 2a**.

R1.7 Figure 2g – I do not believe there is a good reason to include both the corrected and non-corrected data here given the phylogenetic non-independence of your traits. Just include the corrected.

We agree and have removed the non-corrected statistics.

References cited in the review.

-Price S. A., Schmitz L., Oufiero C. E., Eytan R. I., Dornburg A., Smith W. L., Friedman M., Near T. J. and Wainwright P. C. 2014 Two waves of colonization straddling the K–Pg boundary formed the modern reef fish fauna. *Proc. R. Soc. B.*2812014032

-Adams, D. C., 2014, A Generalized K Statistic for Estimating Phylogenetic Signal from Shape and Other High-Dimensional Multivariate Data, *Systematic Biology*, 63, 5, Pages 685–697

-Mitteroecker, P., Collyer, M. L., Adams, D. C., 2024, Exploring Phylogenetic Signal in Multivariate Phenotypes by Maximizing Blomberg's K, *Systematic Biology*,, syae035

Thank you for providing these references. We now cite them in the revised manuscript.

Reviewer #2 (Remarks to the Author):

How niche partitioning contributes to evolutionary radiations is a longstanding interest of evolutionary biology. Whereas the specialization of species that are part of radiations has been relatively well studied in terms of diet and space, much less is known about temporal niche specializations. This manuscript presents a significant advance in this topic. It leverages one of the most classic evolutionary radiations —that of the African Great Lake cichlids— and uses careful behavioral analyses to demonstrate how species throughout the radiation and representing all major lineages, have become diurnal, nocturnal, crepuscular, and cathemeral. It then studies the genetic bases of temporal activity patterns across the species.

The manuscript is well written and the figures are clear. I have some reservations about the interpretations of some findings, particularly in the genetics part.

As a minor note, it would have been nice if the pages and lines were numbered, to make it easier for me to refer to them throughout my review here. I'll try to be as clear as possible as to what I'm referring to, though.

We thank the reviewer for their positive comments on the significance and clarity of our manuscript. We believe we have fully addressed their comments. In particular, we have clarified our genomics methodology to highlight the quality control and filtering methods we used, as well as clarity regarding the gene ontology analysis of genes associated with SNPs. In addition, we have added a discussion of limitations to our study and the interpretation of our results. Specific comments are addressed in detail below and line numbers are now specified.

R2.1 Because temporal activity patterns appear fairly viable, with little phylogenetic signal, it is not clear if the temporal preferences adopted by each species contributed to the radiation or evolved “after” the radiation, without contributing towards speciation. For example, maybe species drifted towards different temporal preferences and sexual selection required that animals of the same species are active at the same times, rather than species ecologically partitioning across the day? I invite the authors to decide whether they can distinguish between the multiple alternatives. At the very least, they should discuss this as a limitation of their results and temper the conclusions in the abstract and throughout the manuscript.

We thank the reviewer for these interesting comments, and we agree that there are multiple interpretations of our data. We cannot explicitly reject the hypothesis that temporal activity patterns evolved through drift, and were subsequently strengthened through sexual selection. However, our observations of species with otherwise similar eco-morphological niches but differing temporal niches, particularly the four sets of species we highlight in **Extended Data Fig. 2b** and **Fig. 2f**, suggest that if these temporal barriers did not exist, those species would occupy the same niche and compete. Future work examining the activity patterns of a larger portion of the radiation would provide further opportunities to test these hypotheses, particularly by allowing a thorough analysis of the temporal dynamics and evolution of these traits through time using phylogenetic approaches. In place of this, we offer an expanded discussion of these limitations and alternative hypotheses in the discussion (including **lines 379-395 & 396-419**). See response to reviewer **R1.3** and **R2.4** for more discussion of this point.

R2.2 The inference of which genes and what gene classes associated with temporal activity patterns are not convincing. First, the methods are not clear as to how SNPs were linked to causal genes. I believe it was based on what gene was closest to the SNP. However that approach is problematic, since the nearest gene is very often not the gene affected by the SNP. For example, the genetic variant that promotes lactase persistence in humans—one of the best documented cases of human adaptation—is in an intron of the neighboring gene (MCM6). There are many other examples like this. This affects the conclusions regarding the outlier genetic variants, which were reported to potentially affect specific genes. More problematic is the gene enrichment analysis. The intergenic regions of neuronal genes are 2–3 times larger than the median intergenic size (<https://doi.org/10.1016/j.neuron.2021.10.014>). This gives more space for variants to be located near neuronal genes, and contribute to fortuitous results in gene enrichment analyses. Other biases as to how genes are classified in databases compound the inaccuracies. Altogether, without functional validations, it is impossible to know which genes are affected by the associated SNPs nor how many of the inferred genes and gene sets represent false positives or false negative results. I recommend removing the gene set analyses reported in Fig 3g and elsewhere and to be more tentative about the genes reported in Fig 3e-f and in the text.

We apologize to the reviewer that our methods regarding association of SNPs and gene enrichment were not clear. We would like to reassure the reviewer that 1) we used a well supported program to link SNPs to genes (snEff). This algorithm is not limited to a nearest neighbour approach, and these associations were not used for GO analysis; 2) we used methods for gene ontology designed specifically for SNP based datasets that account for gene length differences amongst gene sets.

Regarding the association of SNPs to genes, we agree with the reviewer that the functional effect of each SNP may not be directly tied to the nearest gene. We used the default behaviour in snEff to assign genes to each SNP, including multiple genes for many SNPs. For example, the SNP associated with variability in PC1 loadings across species highlighted in **Figure 3d** was assigned to three genes by snEff: it was within the intron of the gene *tpst1l* and downstream of both *tmtops3b* and *crcp*. We chose to highlight *crcp* due to its known function in regulating sleep/wake patterns, rather than either of the other genes, which do not have known roles in sleep or wake states. The location of each SNP, and its relationship to each gene in our dataset are included in **Supplemental Data 3**. We would like to note that these associations were not used in the gene ontology analysis, or any other downstream analysis, other than our own explorations of the data and to provide labels for Manhattan and scatter plots in **Extended Data Figure 7** and **Figure 3**.

Regarding the gene ontology analysis, we used the program SNP2GO (<https://doi.org/10.1534/genetics.113.160341>) to identify enriched ontology terms associated with each set of associated SNPs (for PC1, PC2, and total rest). Importantly, this program takes into account the length of the DNA sequence for each gene, and controls for gene sets that contain genes with longer or more numerous introns. Additionally, it also controls for local genomic effects, including the distribution of SNPs, clustering of genes with similar functions, and heterogeneity in mutation rates across the genome.

We have added a discussion of the limitations of these analyses to the revised manuscript, including a call for further functional work to validate these observations (**line 427**).

R2.3 The genetic analysis also relies on many assumptions that are not acknowledged in the text: for example, it assumes that the same genetic variants affect temporal activity patterns across species (as opposed to being “private” to each species) and therefore that such variants predate speciation or that are flowing across species more recently through hybridization. These and other assumptions should be made explicit in the paper and backed up by the literature or new analyses as necessary. This is for clarity and so readers who are not experts in cichlid genetics can understand the findings and their limitations.

We thank the reviewer for these suggestions. We have added clarification to the text regarding the assumptions taken for the genetics of shared traits in cichlids (**line 256**).

R2.4 There is not enough evidence to conclude that “diurnal and nocturnal activity patterns are facilitated through an intermediate crepuscular state or ‘bridge’” based solely on the existence of diurnal, nocturnal, crepuscular, and cathemeral species. Are the 60 analyzed species not enough for a formal study of evolutionary transitions? Otherwise we do not know if diurnal species are more likely to evolve from nocturnal or crepuscular species or vice versa or if all transitions are equally likely.

We thank the reviewer for this comment, as well as the similar comment raised in **R2.1**, and which is similar to a comment from reviewer #1 (see **R1.3**).

Please see response to reviewer #1 comment **R1.3** for more details. We have now included analysis and interpretation of ancestral reconstructions in our manuscript (**Extended Data Figure 5d-g**) (**lines 231-249 & 587-596**). This analysis suggests that the relationship between diurnal/nocturnal preference and crepuscular activity is maintained in internal nodes, and that the level of crepuscularity a species has is related to the rate of evolution in diurnal/nocturnal activity preferences. We also discuss limitations to this approach, including our limited taxonomic coverage which lacks monophyletic sister species and representation from some early branching clades.

We have added discussion of the limitations of our interpretations of the crepuscular bridge model. Moreover, we now highlight two alternative, but not exclusive hypotheses, on evolutionary transitions between activity patterns in the discussion (**lines 379-395 & 396-419**).

R2.5 The site filtration appears appropriate but I’m surprised there was no variant call filtration. For example, even when a site passes filters, the variant calls can be of low quality in some or most samples. Calls should also be filtered based on GQ and or other quality metrics and then you could remove sites with calls that don’t pass filters in a minimum fraction of samples.

We apologize to the reviewer for any confusion regarding the variant filtration methods we used. As the reviewer points out, we performed extensive site filtration based on read depth, as well as the ability of pseudo-reads to map accurately back onto the genome. We also filtered out INDELS, multi-allelic sites, and variants that were only found in one species.

Importantly, we did not consider any variants that had missing calls in any of the species we examined. Additionally, we also used quality scores from GATK, including the QD metric, which is the quality of a variant call normalized by the read depth at that variant, to remove low quality variants from downstream analysis. Details on these filtration steps are found in the methods section (**line 624**).

R2.6 It was not clear why indels were excluded from the analysis. If the argument is that each of the 39 million variants are mostly independent, then indels are not being tagged by linkage disequilibrium with SNPs and excluding them will bias the results by removing variants that are more likely to affect function than SNPs.

We agree with the reviewer that INDELS may be functionally relevant, and may thus be associated with differences in temporal activity patterns and rest. INDELS could potentially also be examined through similar approaches as ours. However, due to difficulties in the analysis of INDELS compared to SNPs (most importantly higher error rates in their alignment and calling vs SNPs), we chose to focus on SNPs in this study. We also excluded multi-allelic sites, and variants that mapped to unplaced scaffolds. Using the same GWAS pipeline that we used, a similar study from some of our authors has recently identified a functional variant that is a SNP and that controls exploratory behaviour across cichlid species. We have added further clarification on the use of INDELS vs SNPs (**line 264**). Future studies could examine the role of INDELS and other genomic alterations (such as gene duplications) on the traits identified in this study.

Minor comments

R2.7 What does “online tracking” mean? That it was done as the animals were behaving rather than after? As I understand it, this is not relevant to this paper, so could be removed for clarity.

As the reviewer mentions this is not relevant and therefore we have removed any mentions of “online tracking” from the manuscript (**lines 84 & 699**).

R2.8 “PLS scores representing diurnal, nocturnal... were not correlated with either morphology or environment (Fig 2b, EDFig 3)”. Those figures do not show that data.

We apologize for the confusion and the error in the reference to the figures. The reference to **Extended Data Figure 3** should in fact be to **Extended Data Figure 5**. However, the reference to **Figure 2b** is correct. We used the same raw data (measurements of speed across the 24hr period) for both the PC analysis and the PLS analysis. In the methods section we state that PC1 and PC2 for temporal activity patterns which were calculated by PLS represented day-night preference and crepuscularity, and were highly similar to principal components for activity patterns (PC1 and PC2 in Fig. 1 above). In other words, the principal components identified by the PLS analysis were almost identical to the principal components identified by regular PCA. Therefore, the variable “speed” in Figure 2b represents the correlation of these components with the other traits (body morphology, jaw morphology, stable isotopes). We have modified the figure for clarity (**line 192**).

R2.9 “Alleles whose absence was associated with the trait”. I don’t understand this wording. Do they mean that the reference allele was associated with the trait, as opposed to the alternate allele?

This is correct, we have clarified this in the text (**line 286**).

R2.10 “Much of this complexity arises from differential association of HAVs between clades”. I don’t follow the logic that lead to this conclusion.

We have clarified this point in the text to better explain that we observe extensive lineage stratification in the signal (the polygenic loci associated with the traits are different in each clade), but that also there might be loci of strong effect which are ‘private’ to individual clades (**line 293**).

R2.11 Related to my comment above about lack of clarity of how SNPs were associated with genes: “...we annotated all HAVs using SnpEff. We termed such genes...” Which genes? Those that SnpEff annotated based purely on proximity when they were non-coding? How many lead to missense or nonsense or potential splicing effects on specific genes?

We have clarified the text regarding annotation of HAVs and identification of HAGs. In addition, all annotation results for HAVs for each trait are included as csv files in Supplementary Data 3. We have also acknowledged that this approach can lead to false positives (genes which are not affected by the variant in question, **line 302**), and detailed examination is provided for top SNPs (see response to reviewer R2.2 above).

R2.12 I didn’t understand what evidence led to this conclusion: “Temporal activity patterns in cichlids are polygenic, primarily due to differential usage of SNPs across clades.”

Our analyses suggest that many loci are associated with temporal activity patterns across cichlids. However, many loci seem to have alleles that are private to certain subclades (tribes) in the radiation. This could suggest that some loci have strong effects, as seen for exploratory behaviour (Sommer-Trembo et al. 2024), but only within a subset of species. For example, an allele might be associated with diurnality only within the Tropheinei, where it has a strong effect, but not underlie diurnality in Lamprologini, where a different locus, or different combination of loci might be associated with temporal activity patterns. We have now clarified the text (**line 293**)

R2.13 The discussion mentions “modelling studies have also demonstrated that changes in the activity phase of circuits downstream of the clock can explain switches between diurnal and nocturnal behavior.” A lot more work has been done in studying the suprachiasmatic nucleus that controls circadian rhythms and how it differs between diurnal and nocturnal rodents and other mammals. It would be good to cite such work.

We thank the reviewer for this excellent suggestion. We have extended this discussion point and included more key citations to capture more of the circadian and nocturnal/diurnal literature (**line 405**).

R2.14 There are many typos in the Methods section.

We have corrected typos where evident.

R2.15 The density of fish in the husbandry section is not clear. What is a density of “0.5 cm/litre”? Don’t you measure density in number of fish per liter?

We apologize for the confusion. This measurement means a density of 0.5cm of fish body length/litre. Body length is a standard measurement of fish size, and so this takes into account both the number and size of fish per tank. We’ve clarified this in the Fish Husbandry section of the methods (**line 446**).

R2.16 The source of species in the “Behavioural assays” section is repetitive and somewhat inconsistent with the “Husbandry” section.

We have clarified and consolidated the fish species sources to the Fish Husbandry section of the methods (**lines 452 & 479**).

R2.17 What does it mean that “sex of was recorded and used to disaggregate results...”?

This is not relevant for this work, so we have removed it (**line 493**).

R2.18 I don’t understand how the behavior of fish was recorded. It says it was recorded from the front. If they were only recorded from the front, how does movement from the front to the back of the tank get measured?

The reviewer is correct in their understanding that the fish are only viewed from one side (the front). In any one-angle tracking setup, movement along the dimension that is shared with the recording angle is impossible to measure, or can only be roughly approximated by changes in silhouette size. To account for this, the dimensions are typically restricted along that dimension to promote movement that is easily measurable. Our tank design accounts for this, with the narrowest dimension being front-to-back (25 cm). Additionally, in our testing and observations most, if not all, movement periods are dominated by left-right or up-down movement. So while there may be a small underestimation of activity, our setup allows us to recover the large majority of activity. This one-sided top-or side-view method is used by many laboratories due to its ease of use, and largely accurate ability to capture the majority of activity. Examples of one view (side or top) for tracking activity in fish species:

Cavefish: <https://pubmed.ncbi.nlm.nih.gov/30958465/>

Zebrafish: <https://pubmed.ncbi.nlm.nih.gov/38490200/>

Cichlids: <https://pubmed.ncbi.nlm.nih.gov/33658242/>

Sharks: <https://pubmed.ncbi.nlm.nih.gov/32525441/>

R2.19 The usual background subtraction was 1 hr but then later it says that when necessary, background subtraction was increased to 30, 20, or 10 minute periods. This is not clear.

We have made the background calculation, its purpose and the different periods clearer in the text (**lines 513-520**). Briefly, many cichlid species regularly moved the sandy substrate

around, often constructing hills or other structures within timeframes shorter than 1 hour. Using shorter background averaging time frames allowed us to account for these changes in the background.

R2.20 Is “v0.0” a typo for the version of sklearn.decomposition?

Thank you for picking this up, sklearn.decomposition is part of the package scikit-learn which was version 0.24.0. We have corrected this in the text (**line 538**).

R2.21 Is a fixed threshold of 15 mm/s appropriate to quantify rest of all species, given that they vary so much in size? It would be good to show that the accuracy is equivalent across species of different sizes or to adjust thresholds to body size.

We thank the reviewer for this comment. We chose a fixed threshold for calling movement/rest, as there was no correlation between a fish’s size and its maximum speed in our setup (**New Extended Data Figure 2a**). Moreover, the calculated amount of total rest across all species does not change when using a per individual 25% body length threshold instead of a fixed 15mm/s threshold for all species (**New Extended Data Figure 2b**). Given the very poor relationship between body size and fish speed, we therefore opted for the simpler method that is consistently applied across all individuals, and excludes the small movements associated with fin undulations in all species. We have now added these justifications into the main text (**line 138**) and as two new panels in **Extended Data Figure 2**.

R2.22 “We quantified the position of fish, here we scaled the data between 0 and 1, where 0 was the min and 1 was the max”. This is not clear.

We’ve updated the text to be clearer (**line 553**). Fish were tracked within regions of interest (ROIs) from the camera’s field of view, and ROIs were not drawn identically across recording sessions. Briefly we scaled the y position data from 0 to 1 to account for these deviations, where 0 was the minimum and 1 was the maximum fish position. We tracked the centroid position of fish which meant a thinner fish would get closer to the top and bottom edges than at tall-bodied fish. As fish explored the whole arena over the 6 days, using the centroid minimum and maximum allowed us to define what was the top and the bottom for each fish individually.

R2.23 Suppl. Data 4 Table should include the species associate with each GenBank accession number and should also include the coverage after removing duplicates for each accession. Also cite the publication associated with each accession.

We thank the reviewer for this suggestion. These sequences are all from the same publication, and this information was also provided in the original manuscript. However, as requested, we have replicated this information in our own **Supplementary Data 4**.

R2.24 I’m not getting much out of ED Fig 6 regarding genetic associations with behavior. It is difficult to make sense of the patterns, which are more clearly highlighted in Fig 3a,b.

We apologize for the confusion. **Extended Data Figure 6** contains information on the allele associations for total rest and crepuscularity (PC2 values), **Figure 3a-b** contains information of the allele associations for diurnal-nocturnal preference (PC1 values). In the text we describe that we observe similar groups of alleles based on their associations with each behaviour across the different cichlid species tribes. We have added matching colouration to **Extended Data Figure 6** to better illustrate the similarities in patterns.

Reviewer #3 (Remarks to the Author):

Thank you for the opportunity to review this manuscript. In it the authors present a comprehensive study on the understudied role of temporal niche partitioning in the widely investigated adaptive radiation of Lake Tanganyika cichlids. The authors obtained a comprehensive behavioral tracking dataset to monitor activity patterns of individuals from a large number of species over several days. Their data clearly points toward crucial differences across species in activity between light and darkness periods of study, as well as during periods with gradual changes in the light environment. They further integrate their data with readily available ecological and morphological data for these species, and conducted well executed genome-wide association studies that contribute with initial insights on molecular mechanisms underlying changes in daily activity patterns across species. Undeniably, this study provides an important contribution to further our understanding on the role of temporal niche diversification in this important system. I also believe their findings will be influential for future studies evaluating drivers of species diversification in this and other adaptive radiations, as well as across other systems.

R3.1 I have however more reservations than those shown by the authors that their data provides direct evidence that temporal niche partitioning was crucial in the adaptive radiation of these fishes. The experimental design of the behavioral tracking dataset, while impressive, might unfortunately not be enough to capture ecologically-relevant changes in temporal activity across different light environments. I completely understand the practical decision to study activity of individuals while virtually isolated from social activity (with only access to visual and smell cues of very few conspecifics). Yet, the activity patterns observed from different species might be influenced by behavioral differences across these species in how they respond to this asocial situation. I would imagine that the fact that activity patterns across light environments were not correlated with any specific known morphological feature are quite unexpected, and might be indicative that the obtained behavioral data is capturing certain noise around values indicative of only circadian rhythms of these species. In this line, their results from genetic analyses found mostly variants previously associated with regulation of neural function, social behaviours and human disorders clearly associated with responses to social situations. While the authors provided a few examples from species in which observed temporal activity patterns matched expected physiological or ecological adaptations, I think a formal systematic comparison of their categorization with a trait known to vary strongly across diurnal-nocturnal species would strengthen the idea that they are capturing natural temporal activity patterns in their setup. I personally think those related to sensory systems should be ideal for this, as described in some of their examples.

We thank the reviewer for their positive comments, and acknowledge the appreciation of the reviewer for our experimental design and scope, and the potential impact of our manuscript.

We agree with the reviewer that our experimental design, though practical, may limit understanding of the circadian activity patterns of these species under more ecologically-relevant contexts. For example, our setup is missing many factors that are present in the natural context including conspecifics, predators, prey, rocks/shells/plants, temperature variations, and seasonal or monthly cycles in temperature or light. Each of which may be different according to the niche of each species. Further studies in this system

would be necessary for a more complete understanding of these factors on activity patterns across species.

In response to this comment, as well as the comment from Reviewer #1 (see comment **R1.1**), we describe here analyses of the behaviour of example species (including diurnal, nocturnal, and crepuscular species) recorded in their home tanks, in a more natural social environment, and which match the behavioural patterns observed in our reductionist setup (See **Response to Reviewers Figure 1**).

In addition, and as suggested, we have performed morphological analyses of eye size (a sensory system as suggested), as well as body size (see comment **R3.3** below), and their correlation with temporal activity patterns. Information on eye size is readily available for all species and contained in the body morphology data from Ronco et. al 2021. Though we did observe a trend in the data for both nocturnal and cathemeral species to have larger eyes than diurnal or crepuscular species, no significant relationship between either categorical assignments, or between PC1 or PC2 values and eye size was observed across species (**Extended Data Figure 3a-b**). These results suggest that nocturnal, cathemeral, and crepuscular cichlids may have adapted to dim light conditions through a variety of mechanisms, including increased eye size, but also increased lateral line systems, improved olfaction, or even differential expression of opsin genes (Ricci et al, 2023). Additionally, these species live in different environments, including at different depths, which may also affect the evolution of visual and sensory systems (Ricci et al, 2023). We have added discussion of these data to the manuscript (**lines 180-196, Extended Data Figure 3**)).

Finally, and also in response to the other reviewers, we have added more discussion of the limitations of our study and our interpretations to the discussion section, including possible pleiotropic actions of neuronal genes in both temporal activity and social behaviours (**lines 335, 397, & 422**).

Regarding the manuscript's presentation, I found it to be very well-written and easy to comprehend, with visually appealing figures.. As such, I only have a few additional comments regarding methods and potential additions to the manuscript:

R3.2 I think it would be beneficial to provide additional info on how fish were exactly fed during behavioural assays. I am surprised that the analyses did not detect a signal indicating peaks in activity associated with feeding times, as activity patterns typically increase during such periods.

In experimental tanks, fish were fed every other day. The feedings were generally in the mornings, and purposely performed at non-uniform times, with the timing of each feeding recorded. The feedings for all species followed this same general schedule. There are some increases in activity due to feeding evident in the six day traces in **Extended Data Figure 1**. However, this inconsistent feeding schedule was advantageous as the feeding effects were effectively averaged out for the daily activity traces. We have added more information on feeding into the methods (**line 439**).

R3.3 I suggest that the authors extend their discussion on correlations found/not found between behavioral data obtained and physiological measurements. How does their data

align with the existing literature regarding the observed correlation between body size and total rest time?. Also, the lack of correlation between activity and body size seems contractor with a large body of literature in this matter.

We thank the reviewer for this excellent suggestion. We are most familiar with the work describing a negative correlation between body size and total sleep amounts across herbivorous mammals². To test for a similar relationship across cichlid species, we have now included body size in the PLS analysis - we previously only presented relationships between our activity measurements and body morphology. We found a weak but phylogenetically significant negative correlation between body size and total rest across all cichlid species in our dataset (**Figure 2d**). This relationship was significant regardless if we used maximum known body sizes for each species, the maximum size for each species in our data, or the mean size across individuals in our data. Interestingly, this relationship was consistent across diet guilds, suggesting that this relationship is more universal in ectothermic cichlid fish than across endothermic mammals. In addition, we now also provide analysis of the body morphometrics that correlate with total rest across species. Specifically, smaller cichlids or those with elongated bodies had the least rest, and larger or deep-bodied cichlids having the most rest (**Figure 2c**). We have also added discussion of this point to the manuscript (**lines 180-196**).

R3.4 Regarding the methods to capture total rest in tracking data, should the movement threshold not be adjusted based on the size of each individual being tracked?

We thank the reviewer for this comment, which is similar to a comment from Reviewer #2 (R.2.21). In short, the maximum speed and body size of each fish in our experiment are weakly correlated in our dataset (**Extended Data Figure 2a**). Moreover, we tested for differences in total rest between an analysis using a fixed threshold of 15 mm / second or a per-fish 25% body length / second threshold. Analysis of total rest using either threshold results in nearly perfectly correlated total rest values across all species (**Extended Data Figure 2b**). We therefore opted for the simpler method that is consistently applied across all individuals, and eliminates small movements associated with fin undulations in all species.

R3.5 As mentioned before, speed and body size seem to be strongly correlated across fishes. Therefore, I find it somewhat unusual that correlations with body size were calculated using previous data for each species without considering the body size of the individual participants in these experiments within their models.

We have now included individual body size into the analysis discussed in above reviewer point R3.3. In our set up we do not find a strong correlation of max speed (99th percentile of speed) with individual body size (new **Extended Data Figure 2a**). In addition, the correlation we observed between body size and speed is maintained regardless of if we use the size of the fish in our study (new **Figure 2d**), or the maximum size of the species.

Summary:

We thank the reviewers for their useful feedback and comments. We were especially appreciative to hear that it was “exciting to see how the work has progressed” and that they wished to “congratulate [us the authors] once again on an interesting study with a good scientific approach and excellent visualizations”. We were pleased that our previous revisions were so positively received. In this second revision, in response to the further comments, we have:

- 1) Added testing of discrete models for activity pattern evolution as suggested by Reviewer #1 which supports the crepuscular bridge hypothesis.
- 2) Added the social group behaviour experiments and analyses and context from the reviewer response into the main manuscript as suggested by Reviewer #3.
- 3) Addressed further minor comments on clarifications by both reviewers.

Please see below for a detailed point-by-point response to each reviewers' comment.

Reviewer #1 (Remarks to the Author):

Here the authors perform behavioural assays on 60 different cichlid taxa from Lake Tanganyika and then examine the genetic basis for these behaviours.

R1.1 This is the second time I have seen this manuscript, and it is exciting to see how the work has progressed. While almost all my comments were addressed, I would still like to see some empirical data on the ‘crepuscular bridge’ hypothesis. The authors added an ancestral state reconstruction and tested various evolutionary models against their activity data but I have some minor concerns around how these were conducted.

I’m happy to see the authors discuss in their rebuttal the pilot study on the group assessments. I agree that maintaining the health of our study organisms should always be paramount.

We thank the reviewer for their positive comments on our revised manuscript. We address the comment on the “bridge” hypothesis below.

The ancestral state reconstruction gets to some of the questions around activity transitions, although I will caution about over-interpreting results from these analyses. See Holland et al. 2020 for details on this. The larger issue was the evolutionary model analysis – I don’t understand how these data can lead to evidence for or against the ‘crepuscular bridge’ hypothesis. You also don’t assess the fit of your PC activity data to a multi-peak model, whereby peaks correspond to diurnal, crepuscular, nocturnal, cathemeral patterns - this would probably give you the best sense for transition rates. I don’t believe this continuous model fitting analysis adds much to the story and could be removed.

I would however recommend using the routines in Liam Revell’s phytools package – specifically the fitMK family (the help file and included examples are exceptional) – where you can compare the fit of discrete character models. This should give you a better sense for the transition rates among temporal activity patterns. You can then test whether your data fit a model of equal vs. different transition rates. This should allow you to more directly compare rates and make inferences about the ‘crepuscular bridge’ hypothesis.

We previously included analysis of brownian motion models for PC1 and PC2 evolution. This analysis highlighted nodes with higher PC2 values that diverged into daughter species with opposing PC1 values, and nodes with lower PC2 values diverging into daughter species with similar PC1 values.

We thank the reviewer for the alternative suggestion to discretize our continuous measurements of diurnal-nocturnal preference (PC1) and crepuscular preference (PC2) and model them as discrete traits (“multi-peak model”). Therefore, to expand on the ancestral estimations of our continuous traits, and to address these specific points, we have now included analysis based on discrete character states (lines 252-267 and 628-636). We used our assigned activity patterns (“diurnal”, “nocturnal”, “crepuscular”, or “cathemeral”) (**Figure 1**), and compared the performance of multiple discrete trait models that contain parameters representing transitions rates between states. Using the phytools function fitMK (as suggested), we compared the performance of equal-rates (ER), symmetrical rates (SYM), all-rates different (ARD) models, as well as the performance of constrained versions of the SYM and ARD models where direct transitions between diurnal and nocturnal states were not allowed (bridge-only-SYM, bridge-only-ARD). These constrained models mimic the crepuscular/cathemeral bridge hypothesis, allowing us to test the hypothesis against models where direct transitions are allowed. We compared AIC scores between models, which account for the likelihood of the model, while penalising models with a high number of parameters (to avoid the effect of overfitting).

Comparison of AIC scores favoured acceptance of the bridge-only-SYM ($\Delta\text{AIC} = 0$ (AIC = 168.5)) model and rejection of the SYM ($\Delta\text{AIC} = 2$ (AIC = 170.5), ARD ($\Delta\text{AIC} = 9.8$ (AIC = 178.3), and bridge-only-ARD ($\Delta\text{AIC} = 6.5$ (AIC = 175) models ($\Delta\text{AIC} > 2$) (**Extended data Fig. 7h**). Under the bridge-only-SYM model, high transition rates were observed between nocturnal-crepuscular, nocturnal-cathemeral, and diurnal-crepuscular states, with no transitions between diurnal-nocturnal states or between cathemeral-diurnal states (**Extended data Fig. 7i**). These results suggest that cichlids have most likely transitioned between diurnal and nocturnal states through an intermediary crepuscular bridge (**lines 253-266**).

Line by line comments

R1.2 408-412: Can you provide more clarity here? I became lost around the “tissue-specific clocks” line. Is this saying that different tissues and cells can exhibit different rhythms and respond differently at different times of the day?

We apologise for any confusion, and have made these sentences clearer (now **lines 423-432**).

R1.3 495: With the removal of this line, I believe there is no mention of how sex is treated in the manuscript. There should be a note about whether sex was considered to impact activity patterns.

We have added back the mention of sex to the text. We now specify that we did not have enough individuals to quantitatively compare sex for each species (**lines 517-520**). However, we have included a new Extended data Figure showing the daily activity for each species separated by sex (**Extended data Fig. 2**). While most species do not show a difference between the sexes, a few species have some potential differences which could be the basis for future studies.

R1.4 590-594: What would constitute a superior model? Typically set to be around 2 AICc units away from the next best-competing model (Burnham and Anderson 2002). Could present the delta AICc values to make this clear. The fitMK analysis should also include the delta AICc threshold, i.e., that 2-4 units from next competing model is typically considered to demonstrate superior model fit.

Yes, this is correct - we used a delta-AIC threshold of 2 points to differentiate between models. The OU model was 4 points lower than the next closest model for PC1 values, and 10 points lower than the next closest model for PC2 values. We have added clarification of the AIC scores for our continuous models (**lines 621-638**).

In this same section we have added information on the discrete character models and their AIC scores, using the same threshold for demonstrating superior model fit (delta-AIC > 2).

References mentioned in the review.

Burnham KP, Anderson DR (2002) Model selection and multimodel inference: a practical information-theoretic approach, 2nd edn. Springer, New York

Holland, B.R., Ketelaar-Jones, S., O'Mara, A.R. et al. Accuracy of ancestral state reconstruction for non-neutral traits. Sci Rep 10, 7644 (2020).

<https://doi.org/10.1038/s41598-020-64647-4>

Revell L. J. (2024). phytools 2.0: an updated R ecosystem for phylogenetic comparative methods (and other things). PeerJ, 12, e16505. <https://doi.org/10.7717/peerj.16505>

Reviewer #1 (Remarks on code availability):

R1.5 GWAS looks good - code separated into chunks to allow a user to follow each step - file names are informative. Comments present throughout.

ReadME is present - links to preprint.

I did not attempt to run the code.

We are happy to hear that the code is accessible.

Reviewer #3 (Remarks to the Author):

I commend the authors for their revision of the manuscript and congratulate them once again on an interesting study with a good scientific approach and excellent visualizations. This revised version elegantly refines the text without abrupt modifications or major additions, incorporating most of the reviewers' suggestions. In my view, these edits have strengthened the manuscript by enhancing clarity, appropriately tempering the direct conclusions drawn from the findings, and clearly distinguishing speculative aspects in the discussion.

I believe there are still areas where additional improvements could further refine the manuscript. Below, I outline my comments, focusing on the authors' responses to my previous feedback as well as revisions prompted by other reviewers' suggestions.

We thank the reviewer for their positive comments on the revised manuscript, and hope to have now fully satisfied the reviewer.

In addition to minor comments that I consider sufficiently addressed, I have grouped my previous suggestions for improvement into three main categories:

3.1. The relevance of testing species in isolation.

In response to Reviewer 1's suggestions, the authors present new data (included only in the response letter) on activity patterns in three species studied in a more ecologically relevant social context. Given that current editorial policies make response letters publicly available, I understand the authors' rationale for keeping this data separate. However, I respectfully disagree with their decision to exclude it from the manuscript.

This information is important for contextualizing the study's methods and understanding its limitations. As I understand it, the feasibility of conducting experiments in a fully naturalistic setup is constrained by logistical challenges and the difficulty of developing a unified approach for species with significant biological differences. The authors' previous data suggest that individual activity patterns in isolation are representative of those in ecologically relevant conditions. However, the response letter lacks details on species selection criteria and key methodological aspects of data collection. If space allows, I strongly recommend incorporating this information into the main text or extended data.

While I acknowledge the authors' reasoning, I remain unconvinced that testing fish in isolation does not introduce noise affecting observed activity patterns and the genetic variants associated with them. The addition of the term "social behavior" when discussing the genetic findings does not, in my view, sufficiently acknowledge this limitation.

We appreciate your interest in the data and given your request we have added this data to the manuscript (**lines 98-101** and **521-529** in methods, new **Extended Data Fig. 3**). This includes further details on the species selection and methodology of data collection as requested.

We agree with the reviewer that testing the fish in isolation could affect our interpretation of the results. However, as previously noted and appreciated by the reviewers, this reductionist assay was a practical decision that enabled higher throughput of animals and more precise measurement of behaviour than what is currently feasible for assaying populations in naturalistic settings. Despite this limitation, we believe, and were happy to hear that all three reviewers found that our study offers valuable insights into the flexibility of activity patterns across Lake Tanganyika cichlids. To further address the possibility of social behaviour affecting the genetic signatures we identified, we have expanded discussion of these limitations to the discussion section (**lines 450-454**). In addition, we have also added discussion of several references that provide evidence (based on observations of feeding) of nocturnal activity patterns in three species (Neotoa, Neotre, Auldew), and crepuscular/diurnal activity patterns in three species (Neobri, Neopul, and Neosav), which agree with our results for these same species (**lines 95-98**). With these editions we feel confident that our reductionist assay provides valuable, accurate, and ecologically relevant information on cichlid temporal activity patterns.

R3.2. Further comparisons of morphological traits between diurnal and nocturnal species.

The inclusion of correlations between morphological traits and temporal activity patterns is well executed and a valuable addition to the manuscript.

We thank the reviewers for the suggestions to add these interesting analyses and agree that they have been a valuable addition.

R3.3. Considerations for tracking data validity in species with varying morphology and feeding behaviors.

I find this concern adequately addressed with the incorporation of an alternative approach to estimate body size and a clearer description of feeding patterns during testing. I find concerns additionally raised by reviewer 2 in this regard are also adequately addressed.

Additionally, I have some comments based on responses to other reviewers. I found that their comments aligned with mine, and I appreciate the authors' efforts to revise the manuscript accordingly.

All reviewers emphasized the need for a more nuanced presentation of results related to the timeline of temporal activity adaptations and transitions between behavioral types. While I am not an expert in the specific analytical methods suggested by the other reviewers, I believe the authors have addressed these concerns well by incorporating phylogenetic information and refining their interpretations throughout the manuscript.

We thank all of the reviewers for these suggestions and are happy to hear that we have sufficiently addressed their concerns.

R3.4 Regarding the analyses linking genes and gene categories to temporal activity patterns, I did not raise concerns in my initial review, but I now provide additional comments in light of the responses to Reviewer 2 and my own experience with similar analyses:

- SNP-gene associations: The authors apply standard methods using default settings, but I agree that these should be more explicitly stated in the manuscript to ensure clarity for readers unfamiliar with snpEff or SNP2GO. Specifically, it should be clear how genes and GO terms are assigned to SNPs and when these assignments are used for downstream analyses. I was unaware that neuronal genes tend to have larger intergenic regions than the median, but I do not believe this introduces a bias in the authors' analyses, which focus on overall associations with gene types and functional categories. The revised manuscript appropriately clarifies that no causal links can be inferred from these findings.

We are glad to hear that the reviewer agrees with our methodology and approaches.

- INDEL analyses: In response to Reviewer 2 and in the manuscript, the authors partially justify not including INDEL analyses by stating that such analyses are increasingly difficult. While I did not find the absence of INDEL analyses problematic given the study's broader focus, I note that recent methodological advances have made these analyses more accessible.

We look forward to using new methodology in future studies to examine the influence of INDELs on behavioural phenotypes in cichlids.

- Tests of allele stratification and behavioral phenotypes: There appears to be a misunderstanding regarding the statistical validation of allele distributions between clades.

The authors claim significant differences in allele stratification (line 293), but as far as I can tell, these results are not based on formal statistical tests. If such tests have been performed, the methods and results should clarify this. If not, I recommend conducting the appropriate statistical tests.

We have clarified that this result does not represent significance from a statistical test (line 312).

Overall, I appreciate the authors' revisions and believe these final clarifications would further strengthen the manuscript.

We thank the reviewers for their valuable comments that have helped us strengthen and refine our manuscript.

Reviewer #1 (Remarks to the Author):

Here the authors perform behavioural assays on 60 different cichlid taxa from Lake Tanganyika and then examine the genetic basis for these behaviours.

This is the third time I have seen this manuscript, and the authors have made some excellent additions to their paper. I am happy to see an expansion of the model fitting analysis and the incorporation of the pilot study to the main text.

I have a few minor comments for the authors that I list below.

Line 253-266 and Extended data figure 7h. You cite an AIC score of 2 units to distinguish a model from other competing models – the ER model (AIC = 168.35) and Bridge SYM model (AIC = 168.47) exhibit similar support. You need to add a caveat in the main text that the ER model also garnered high support alongside the Bridge SYM model. Similarly, in panel h of Extended data figure 7 the delta AIC for the ER model should be 0 rather than -0.12 as this model exhibited the lowest AIC score, and you should subtract this AIC value from the other model AIC values to gain the delta AIC (i.e., Bridge SYM would be $\Delta\text{AIC} = 0.12$ etc.).

The statement of results in the main text now also includes the results of the ER model (line 260). Additionally, we have added clarification of this point to the figure legend and made the proposed changes to the table as suggested.

Line 431. Add a definition for 'zeitgebers' given the broader readership of this journal.

Done (line XXX).

Line 745. Please add new scripts generated as part of these reviews to GitHub to aid reproducibility. <https://github.com/annnic/cichlid-analysis>

We have made sure to push all new scripts and modifications to the public Github, and we thank the reviewer for this reminder, and apologise that it wasn't done before re-submission!

Reviewer #1 (Remarks on code availability):

As discussed in the Comments to Author section, additional analyses were added to the manuscript and should be included in the GitHub repository. In particular -> <https://github.com/annnic/cichlid-analysis>

Done (see above).